# Antimicrobial Activity of Calixarenes and Related Macrocycles

**DOI:** 10.3390/molecules25215145

**Published:** 2020-11-05

**Authors:** Dmitriy N. Shurpik, Pavel L. Padnya, Ivan I. Stoikov, Peter J. Cragg

**Affiliations:** 1A.M. Butlerov Chemical Institute, Kazan Federal University, Kremlyovskaya St, 18, 420008 Kazan, Russia; dnshurpik@mail.ru (D.N.S.); padnya.ksu@gmail.com (P.L.P.); ivan.stoikov@mail.ru (I.I.S.); 2School of Pharmacy and Biomolecular Sciences, University of Brighton, Huxley Building, Moulsecoomb, Brighton, East Sussex BN2 4GJ, UK

**Keywords:** calixarene, resorcinarene, pillararene, antibiotic, fungicide, biofilm inhibition

## Abstract

Calixarenes and related macrocycles have been shown to have antimicrobial effects since the 1950s. This review highlights the antimicrobial properties of almost 200 calixarenes, resorcinarenes, and pillararenes acting as prodrugs, drug delivery agents, and inhibitors of biofilm formation. A particularly important development in recent years has been the use of macrocycles with substituents terminating in sugars as biofilm inhibitors through their interactions with lectins. Although many examples exist where calixarenes encapsulate, or incorporate, antimicrobial drugs, one of the main factors to emerge is the ability of functionalized macrocycles to engage in multivalent interactions with proteins, and thus inhibit cellular aggregation.

## 1. Introduction

Calixarenes have found an enormous range of applications since their first discovery and later exploitation by researchers worldwide [1]. Their ability to complex small molecules, either within each individual macrocycle or through aggregated nanostructures, has opened up avenues for drug delivery. Functionalization, though upper and lower rims, offers up the possibility of their use as inhibitors of biochemical processes or as prodrugs with the active substituents released in response to external stimuli. Some useful summaries of the broader biological and biochemical effects of water-soluble calixarenes [2,3] and pillararenes [4] have been published.

The use of calixarenes and related macrocycles as antimicrobial agents dates back to the 1950s and Cornforth’s work on *Macrocyclon* (Figure 1) [5]. Since his original antitubercular study, the biological effects of calixarenes and their relatives, the resorcinarenes and pillararenes, have become more widely appreciated. The antimicrobial impact of calixarenes was first assessed in 2002 by Regnouf-de-Vains [6] and has been followed by subsequent reviews [7,8,9]. Herein, we update the field of antimicrobial calixarenes and related macrocycles with the most recent advances.

Before embarking on a survey of these compounds’ antimicrobial effects, several points need to be considered. There are three main modes of action by which macrocycles can induce a biological response. Individual calixarenes may do so through, for example, insertion in a cell membrane to destroy its integrity, or by acting as a prodrug, incorporating a substituent that is cleaved from its parent macrocycle to become biologically active. Macrocycles, or aggregates of macrocycles, can deliver single drug molecules by transporting them in their central cavities, releasing them in response to an external stimulus. Finally, there is the possibility of disrupting the biofilms formed by colonies of bacteria when adhering to surfaces.

In all these cases, the solubility of the calixarene, or its complex, must be considered. If it is very hydrophobic, its application, except perhaps as a part of a topical cream, will be limited. Lack of aqueous solubility will make administration difficult. Aqueous solubility is one aspect of Lipinski’s ‘rule of five’, which states that an orally active drug should have no more than five hydrogen bond donors and 10 hydrogen bond acceptors, a molecular mass below 500 Da, and an octanol-water partition coefficient below five [10]. In practice, not all of these parameters need to be met, but it is a useful rule of thumb, particularly when considering macrocycles with high molecular masses.

Once a calixarene or its complex has been shown to be ‘druggable’, it needs to reach its target. A water-soluble calix[4]arene with quaternary ammonium groups on the upper rim and a fluorophore on the lower rim, shown in Figure 2, was used by Matthews, Mueller, and colleagues to probe calixarene transport into cells [11]. Using fluorescence confocal microscopy, it was possible to show that the molecules were not transported across the membrane by endocytosis, nor did they appear to traverse the membrane between lipid raft domains. The authors posit that other, non-specific processes must be responsible as the macrocycles were found to be retained in the cytoplasm and unable to enter the cell nuclei.

One final thought must be given to toxicity. An ideal antimicrobial will display specific activity against one species of bacteria, while having no effects on any other organisms, whether they be humans being treated for an infection or the diversity of animal and plant life that may exist in a field where crops are sprayed. An encouraging study into the toxicity and biodistribution of *p*-sulfonatocalix[4]arene in mice by Coleman and colleagues using a ^34^S-labelled derivative found no toxic effects in mice up to 100 mg kg^−1^ of body weight (Figure 2) [12]. One animal was given a dose of 400 mg kg^−1^ and died within an hour, whereas 14 others, receiving doses of 40 or 100 mg kg^−1^, were outwardly unaffected. Biodistribution was found in all major organs, but particularly the lungs, as well as in blood and plasma. At both 40 and 100 mg kg^−1^, very little was found in muscle tissue and none was detected in the brain. A majority of the 40 mg kg^−1^ dose had been cleared through the urine within an hour.

Taken together, the toxicity data and macrocycles’ cellular destination indicate that a mechanism, yet to be determined, exists by which they can enter cells; however, they are not in themselves harmful.

## 2. Molecular Prodrugs and Drug Delivery Agents

Phenol-based macrocycles provide a rigid scaffold to which numerous substituents can be appended through lower rim substitution reactions. The regiochemistry of these reactions can be controlled by numerous well-known techniques to maximize the distance between substituents, in alternate conformations, or make use of the macrocyclic cavity to preorganize the spatial relationships between substituents in a cone conformer. If the substituents are pharmacologically active in their own right, the macrocycle becomes the agent that delivers the prodrug to a destination within the target microbe. Once at the destination, local conditions, typically pH or enzyme action, initiate bond cleavage to release the active drug while the macrocycle is either chemically degraded or excreted. Alternatively, the calixarene essentially acts as a vector by delivering its drug-derived substituents to their destination while remaining attached throughout. The benefits of such an approach are that the solubility of the macrocycle can enhance the bioavailability of the drug and that multiple copies of that drug can be delivered to the same cellular site. This latter effect is an example of multivalency and is important in other antimicrobial applications of calixarenes, notably in biofilm inhibition.

It is often only through conjugation to a carrier molecule that a drug can be delivered to its target either because of solubility issues, which are addressed by tuning the carrier’s physical properties to those of its biological target, or through the affinity of the carrier for a particular feature of the target such as a phospholipid membrane or protein recognition site. This ‘magic bullet’ approach has long been the goal of medicinal chemists, particularly when the therapeutic agents involved are highly toxic [13].

### 2.1. Incorporation of Antibiotic Motifs

The group of Regnouf-de-Vains has investigated the effects of introducing classical antibiotic motifs into calixarenes over research covering two decades. Calix[4]arenes have been used as a platform for drug delivery through the incorporation of penicillin core moieties as lower rim substituents (Figure 3). The lower rim of 4-*t*-butylcalix[4]arene was functionalised in the 1,3-positions with acid groups, and then further reacted with *N*-hydroxysuccinamide in the presence of dicyclohexylcarbodiimide to form a 1,3-diester. The β-lactams’ carboxylic acids were protected as pivavoyloxymethyl esters and reacted with the calixarene diester to form diamide **1** [14]. Although antimicrobial data were not published, this route led the way to an analogue, **2**, in which penicillin V was appended to the calixarene [15] and tested against Gram-positive and Gram-negative bacteria [16]. The group also prepared a nalidixic acid delivering prodrug, **3**. Nalidixic acid is a bacteriostatic agent that works by interfering in the DNA replication mechanism rather than as a biocide, which kills the host. Reaction of the sodium salt of nalidixic acid with dibromopropane followed by *O*-alkylation of 4-*t*-butylcalix[4]arene in the 1,3-positions gave the disubstituted calixarene in 65% yield [17]. A fourth derivative incorporating nalidixic acid and penicillin V substituents on opposite rings, **4**, was prepared and tested for antimicrobial activity [16].

Given the poor aqueous solubilities of these 4-*t*-butylcalix[4]arene derivatives, antimicrobial disc-diffusion tests were done in dimethyl sulfoxide with the pure solvent as a control. Only Gram-negative reference strain *Pseudomonas aeruginosa* ATCC 27853 appeared to be affected by DMSO and, as the same degree of diffusion was seen for all compounds, it was determined that the antimicrobial effects in all cases were due to the solvent alone. No significant activity was found for Gram-positive reference strains, *Staphylococcus aureus* ATCC 29213 and *Enterococcus faecalis* ATCC 29212, as well as against Gram-negative reference strain *E. coli* ATCC 25922. Only calixarenes **2** and **4**, with two penicillin V or one penicillin V and one nalidixic acid subunit, had any significant inhibitory effect, and then only against *S. aureus* ATCC 25923 [11].

Considering the solubility issues with 4-*t*-butylcalix[4]arene, the group also explored a 4-guanidinoethylcalix[4]arene derivative, **5**, prepared through the reaction of Boc-triflylguanidine with 4-ethylaminocalix[4]arene (Figure 4) [18,19]. A monomeric analogue of the calixarene was synthesized by the same method to assess the importance of the macrocyclic scaffold. To determine mammalian toxicity, MTT assays were performed for both compounds using human embryonic lung fibroblast (MRC-5) cells. Selectivity indices of IC_50_/minimum inhibitory concentration (MIC) showed that the monomer was as toxic as it was active, whereas the calixarene had remarkably low toxicity. Both disc diffusion and minimum inhibitory concentration tests were carried out on Gram-negative *E*. *coli* ATCC 25922 and *P*. *aeruginosa* ATCC 27853, and Gram-positive *S*. *aureus* ATCC 25923 and *E*. *faecalis* ATCC 29212 bacteria. At 18 h, the monomer required concentrations of 512 μg mL^−1^ or higher to have any activity at all, whereas the calix[4]arene was active at 16 μg mL^−1^ against *E. coli*, *S. aureus,* and *E. faecalis*, and at 64 μg mL^−1^ against *P. aeruginosa* [20]. A later study extended the range of bacteria tested to include a second *S. aureus* strain, ATCC 29213, which gave identical results to the ATCC 25923 strain, and clinical isolates of penicillinase-producing *E. coli*, methicillin-sensitive *S. aureus* (MRSA) (mecA gene), *E. faecium* (vanA gene) and (vanB gene), and *P. aeruginosa* (overexpression of efflux pumps) [21]. The effects of reference antibiotics amoxicillin, oxacillin, vancomycin, and ticarcillin were also determined for comparison. The calixarene’s efficacy was unchanged against the clinical isolates, indicating that its mode of action was unaffected by the resistance mechanisms commonly employed by antibiotic-resistant bacteria. The authors comment that, in presenting its four cationic groups co-facially, the calixarene mimics the cytoplasmic perturbation mechanism observed for cation-rich polypeptides. This is an example of multivalency, which is also key to the activities of biofilm-inhibiting calixarenes, discussed later. In 2010, the group published a more comprehensive assessment of the antimicrobial activity of the cationic calixarene against 69 clinical isolates compared with two antiseptic cationic compounds, chlorhexidine and hexamidine [22]. Interestingly, in another study, the group found a 4-*t*-butylcalix[4]arene with *n*-propylguanidinum lower rim substituents, **6**, was most effective in the *1,3-alt* conformation [23]. Tests against Gram-positive *E. faecalis* and two strains of *S. aureus* and Gram-negative *E. coli* and *P. aeruginosa*, together with reference and isoniazid (INH)-resistant *Mycobacterium tuberculosis*, showed that the *cone* conformer had MICs of 8 μg mL^−1^ and 9.5 μg mL^−1^ against *E. faecalis* and INH-resistant *M. tuberculosis*, respectively, but otherwise had MICs between 32 and 128 μg mL^−1^. By contrast, the *1,3-alt* conformation had values below 10 μg mL^−1^, apart from *P. aeruginosa*, for which it was 16 μg mL^−1^. This conformer was particularly effective against INH-resistant *M. tuberculosis*, with an MIC of 1.2 μg mL^−1^.

Another water-soluble derivative, an ethylaminocalix[4]arene with a single nalidixic substituent on the lower rim, **7**, was prepared through *tert*-butyloxycarbonyl-protection of the amines followed by alkylation with bromopropylnalidixate and upper rim deprotection (Figure 4) [24]. In biological media, the prodrug decomposes and releases about 30% of the nalidixic acid over the first 24 h, as assessed by HPLC. An inhibitory effect on the growth of Gram-negative *E*. *coli* ATCC 25922 (MIC 25 μg mL^−1^) and Gram-positive *S*. *aureus* ATCC 25923 and 29213 (MIC 70 μg mL^−1^ for both) was found, but Gram-negative *P*. *aeruginosa* ATCC 29212 and Gram-positive *E*. *faecalis* ATCC 27853 (MIC 140 μg mL^−1^ for both) were relatively unaffected. An analogue without the nalidixic acid, **8**, was broadly ineffective. The parent ethylaminocalix[4]arene was also tested against the same strains and shown to have activity against *E*. *coli* (MIC 4 μg mL^−1^) and both *S*. *aureus* strains (MIC 8 μg mL^−1^) and, to some degree, *E*. *faecalis* and *P*. *aeruginosa* (MIC 32 μg mL^−1^) [25]. This is an apparent improvement over the activity of the naldixic acid derivative, but similar to that of 4-guanidinoethylcalix[4]arene.

Pur and Dillmaghani introduced 6-aminopenicillanic acid substituents to calix[4]arene via esterification at either the lower rim, **9**, or upper rim, **10**, to produce ‘calixpenams’, followed by oxidation to their sulfoxide derivatives and finally through ring expansion and sulfur insertion to cephalosporin-containing analogues, or ‘calixcephems’, **11** and **12** (Figure 5). The calixpenams’ MICs against *S. pyogenes* ATCC 19615, *Streptococcus agalactiae* ATCC 12386, and *Streptococcus pneumoniae* ATCC 49619 were found to be five to six times lower than penicillin V or X [26]. Tests against five strains of β-lactamase-producing or -non-producing methicillin-sensitive *S. aureus* were determined [27]. The calixpenams were more effective against methicillin-sensitive *S*. *aureus* (MSSA) β-lactamase (−) than MSSA β-lactamase (+), whereas the calixcephems showed broad antibiotic activity. In all cases, the calixarenes were between 6 and 10 times as effective as their penicillin or cephalosporin analogues. The synergistic effect of the four substituents was suggested as one reason for the enhanced activity.

### 2.2. Incorporation of Oxazole, Thiadiazole, and Bithiazole Motifs

Menon and colleagues prepared a library of nine 4-*t*-butylcalix[4]arenes, **13** to **21**, with 1,3,4-oxadiazole and 1,3,4-thiadiazole derivatives of isoniazid, nicotinic acid, benzoic acid, and *cis*-cinnamic acid (Figure 6) [28]. These, along with the unattached oxadiazole and thiadiazole derivatives, were assessed for activity against *S. aureus* MTCC 96, *S. pyogenes* MTCC 442, *E. coli* MTCC 443, *P. aeruginosa* MTCC 1688, and fungal species *Candida albicans* MTCC 227 and *Aspergillus clavatus* MTCC 1323. While the zones of inhibition for all compounds in DMSO were similar (8–19 mm), marginally better than ampicillin, there was more variability against the two fungal species. The oxadiazole and thiadiazole substituents were less active (*C. albicans* 6.5 to 16.2 mm, *A. clavatus* 11.5 to 16.4 mm) than when appended to calixarenes (*C. albicans* 9.0 to 23.3 mm, *A. clavatus* 11.3 to 24.3 mm), but none are quite as effective as griseofulvin (24 mm). Separately, the authors assessed the compounds’ inhibitory effects against *M. tuberculosis* H37Rv and found that calixarenes with isoniazid substituents, **13** and **15**, displayed greater than 90% inhibition, significantly higher than the 71 to 82% range for the substituents alone.

Gezelbash and Dilmaghani subsequently reported four 4-*t*-butylcalix[4]arene-based aryl-oxadiazole derivatives, **22** to **25** (Figure 7), in which the substituents were linked by ethanethioate moieties in the 1- and 3-rings [29]. These, along with 14, previously reported by Menon [21], were assessed for their effects against *E. coli* and *Aspergillus fumigatus*. The furyl-containing derivative was found to be the most effective against *A. fumigatus*, whereas the 3-nitrophenyl derivative had broad spectrum activity.

A family of six water-soluble calixarenes, **26** to **31** shown in Figure 8, incorporating 4-methyl-2,2′-bithiazole lower rim substituents on the 1- and 3-rings, was reported by Regnouf-de-Vains and colleagues [30]. Their effects against cells infected with HIV were determined and, compared with three parent derivatives with sulfonate, **32**; carboxylate, **33**; and phosphonate, **34**, upper rim substituents. While none appeared cytotoxic to uninfected cells, only a 4-sulfonatocalix[4]arene derivative displayed any significant anti-HIV activity.

### 2.3. Functionalized Nanoparticles

An alternative approach to delivering drugs as macrocyclic substituents is to utilize the calixarenes’ central cavity to act as a controlled release site for drug guests. Coleman and colleagues used six water-soluble calixarenes, **32** and **35** to **39** (Figure 9), to cap silver nanoparticles (AgNPs) and subsequently to bind chlorhexidine and gentamycin with the presumption that the surface decorated AgNPs could be used to deliver the drugs in high concentrations [31]. In related work, a further six 4-sulfonatocalixarene derivatives, **40** to **45**, were used to cap AgNPs and to inhibit the growth of *Bacillus subtilis* 168 and *E. coli* [32]. No inhibition was observed for *E. coli*, however, four derivatives were effective against *B. subtilis*. The parent sulfonatocalix[4]-, -[6]-, and -[8]arenes together with sulfontocalix[6]arene with *O*-propylsulfonate lower rim substituents all retarded the growth to a maximum optical density from 16 to 18 h.

### 2.4. Drug-Delivering Calixarenes

Wheate and colleagues investigated the potential for 4-sufonatocalix[8]arene, **41** (Figure 9), to act as a vehicle for the antibiotics isoniazid and ciprofloxacin [33]. In human kidney cells (HEK-293), the isoniazid complex was shown to increase uptake owing to the solubilizing effects of the macrocycle by 35%, although the ciprofloxacin complex showed no effects up to its solubility limit of 185 μg mL^−1^. The MICs of the complexes against *E. coli*, *S. aureus*, *E. faecalis*, and *P. aeruginosa* were unchanged from the values of the drugs alone, however, the authors note that the potential for the large calix[8]arenes to bind two different drugs simultaneously warranted further investigation.

A polycationic calix[4]arene derivative, **46** (Figure 9), was prepared by Nostro and colleagues, through the reaction of *N*-methyldiethanolamine on tetra-propoxy-4-chloromethylcalix[4]arene, to deliver ofloxacin, chloramphenicol, and tetracycline [34]. The calixarene alone had a similar profile to chloramphenicol against *S. aureus* ATCC 6538, but its MIC was lower by a factor of two against MRSA 15, MRSE 17, and *P. aeruginosa* isolate 1. It was lower by a factor of four against *S. epidermidis* ATCC 35984, but performed less well against *P. aeruginosa* ATCC 9027. The authors speculate that the polycationic nature of the macrocycle may have a disruptive effect on the membrane structure of Gram-negative bacteria in a similar manner to the guanidinium derivatives discussed above.

### 2.5. Metal-Binding Calixarenes

The use of calixarenes as cation-releasing platforms was investigated by Memon and colleagues, who used a diamide calixarene derivative, **47** (Figure 10), to complex iron(III) [35] and copper(II) [36]. Iron(III) was found to bind in a 1:1 ratio to the calixarene by a spectroscopic Job’s plot and copper(II) in a 2:1 ratio by the same method. Tests against bacteria and fungi found that the calixarene had MIC values between 1.5 and 3 μg mL^−1^, whereas the iron(III) complex had a value of 0.37 μg mL^−1^ against *S. albus*, *E. coli*, and the fungal strain *Rhizopus stolonifera*. The copper(II) complex had the same MIC for *E. coli*, but was slightly less active against *S. albus* and *R. stolonifera*, with an MIC of 0.75 μg mL^−1^.

Yilmaz and colleagues prepared a number of calixarene diamides, **48** to **51**, designed to bind copper with a 1:1 stoichiometry (Figure 10) [37]. The authors propose that copper binds as Cu^2+^ in a slightly distorted square planar geometry, based on spectroscopic data, although whether the heteroatoms in the lower rim substituents are involved in this was not considered. Their antibacterial and antifungal properties were assessed against four Gram-positive, four Gram-negative, and two fungal strains. Quite surprisingly, given the antimicrobial activities of other calixarene-metal complexes, no antimicrobial activity was seen for Cu∙**50** and Cu∙**51**, even at 10,000 μg mL^−1^. Metal-free macrocycles **48** and **50** displayed some activity against *B. subtilis*, *B. cereus*, and *E. coli*, but at best, this was in the range of 40 to 160 μg mL^−1^. This suggests that it is the nature of the substituent, and not the bound copper, that imparts antimicrobial activity.

Shaabani and colleagues prepared a thiosemicarbazide functionalized calixarene, **52** (Figure 10), and investigated its effects, and those of its transition metal complexes, on *S. aureus* ATCC 29213, *B. subtilis* ATCC 6633, *E. coli* ATCC 25922, *P. aeruginosa* ATCC 27853, *C. albicans* ATCC 10231, and *C. glabrata* ATCC 2001 [38]. The tests revealed a higher antibacterial activity against Gram-positive *B. subtilis*, with an MIC of 31.25 μg mL^−1^, than against Gram-negative *E. coli*, with an MIC of 250 μg mL^−1^, and *P. aeruginosa*, with an MIC of 62.5 μg mL^−1^. The metal complexes generally showed enhanced activity against *E. coli*, with the greatest effects seen for the nickel(II) and zinc(II) complexes, with MICs of 62.5 μg mL^−1^ and 31.25 μg mL^−1^, respectively.

The Shaabani group extended the thiosemicarbazide motif through reaction with salicylaldehyde to give 2-hydroxybenzeledene-thiosemicarbazone **53** (Figure 10) [39]. Reaction with the appropriate metal nitrate salts gave Co∙**53**, Cu∙**53**, Ni∙**53**, and Zn∙**53**, which, along with the parent calixarene, were assessed for their antibacterial effects on *S. aureus* ATCC 29213, *B. subtilis* ATCC 6633, *E. coli* ATCC 25922, and *P. aeruginosa* ATCC 27853, together with their antifungal activity on *C. albicans* ATCC 10231 and *Candida glabrata* ATCC 2001. All had MICs of 31.25 μg mL^−1^ against *B. subtilis*, *E. coli*, and *P. aeruginosa*, but only Cu∙**53** had any activity against *S. aureus,* with an MIC of 31.25 μg mL^−1^. Only **53** and Co∙**53** had any antifungal effects, and then only against *C. albicans*, with MICs of 31.25 μg mL^−1^ and 31.25 μg mL^−1^, respectively. Nevertheless, the authors note that this level of activity is between two- and eightfold better than their analogs prepared from **52**.

Ray, Deolalkar, and Desai reported the synthesis of calixarene-salen hybrid corand macrocycles **54**–**58** together with their silver complexes (Figure 10) [40]. The antibacterial properties of the silver complexes were assessed against *E. coli* MTCC 433, *P. aeruginosa* MTCC 1688, *Salmonella typhi* MTCC 98, *S. aureus* MTCC 96, *S. pyogenus* MTCC 442, and *B. subtilis* MTCC 441. Their antifungal effects on *C. albicans* MTCC 227 were also investigated. These tests showed similar antibacterial effects to ampicillin, however, Ag∙**56** and Ag∙**57** were much more effective against *S. typhi* (MIC of 80.7 μg mL^−1^ and 48.8 μg mL^−1^, respectively, vs. 100 μg mL^−1^ for ampicillin), and all were better than ampicillin against *S. pyogenes*. Ag∙**58** was significantly more potent than ampicillin against all strains tested. Of particular note was its activity against *E. coli* (MIC of 49.5 μg mL^−1^ vs. 100 μg mL^−1^ for ampicillin and 24.8 μg mL^−1^ for ciproflaxin), *S. aureus* (MIC of 49.5 μg mL^−1^ vs. 250 μg mL^−1^ for ampicillin or 50.0 μg mL^−1^ for ciproflaxin), and *S. pyogenus* (MIC of 79.3 μg mL^−1^ vs. 100 μg mL^−1^ for ampicillin and ciproflaxin). All silver complexes were better fungicides than griseofulvin when tested against *C. albicans*. Ag∙**55** and Ag∙**56** were the most active, with minimal fungicidal concentrations (MFCs) of 384.1 μg mL^−1^ and 405.2 μg mL^−1^, respectively, versus 500 μg mL^−1^ for griseofulvin.

### 2.6. Sulfonamide-Containing Calixarenes

The use of sulfonamides as bacteriostatic antibiotics predates treatment with penicillin by a decade, which makes them potentially effective when introduced as functional groups to calixarenes. Hamid and colleagues used upper rim diazotization to introduce sulfonamides, **59** and **60**, and other potentially therapeutic moieties, **61** to **63**, to calix[4]arenes (Figure 11) [41]. The compounds were screened against *B. subtilis*, *S. aureus*, MRSA, *S. epidermidis*, *E. faecalis*, *E. coli*, *P. aeruginosa*, *C. albicans*, and *S. cerevisiae*. Compound **59** showed good inhibition against *S. epidermidis* and *S. aureus*, both at 7.8 μg mL^−1^, as well as MRSA and *B. subtilis*, both at 15.6 μg mL^−1^. It was also active against Gram-negative *P. aeruginosa*, with an MIC of 15.6 μg mL^−1^, and the fungal strain *C. albicans*, with an MIC of 62.5 μg mL^−1^. Activity was greatest against Gram-positive strains, with **60** having the lowest MIC values against MRSA of 0.97 μg mL^−1^ and *B. subtilis* of 0.97 μg mL^−1^. It also showed good inhibition against *S. aureus* with an MIC of 3.9 μg mL^−1^, *S. epidermidis* at 15.6 μg mL^−1^, and *E. faecalis* at 3.9 μg mL^−1^. The tetrasubstituted derivatives **62** and **63** had limited inhibitory activity, which was explained by docking studies on neuraminidase- and penicillin-binding protein receptors. A single substituent was able to bind to the active site, while the macrocycle interacted with a number of neighboring protein side chains. By comparison, the tetrasubstituted derivatives were too sterically crowded for a single substituent to approach the binding sites.

### 2.7. Antibiotic Pillar[5]arenes

Moving away from the calixarene framework, Notti and colleagues utilized the water-soluble deca-carboxylatopillar[5]arene, **64** (Figure 12), as a carrier for the aminoglycosidic antibiotic amikacin and demonstrated that, where the macrocycle:drug ratio was 2:1 or greater, no antibiotic effects were observed against *S. aureus* ATCC 29213 based on the density of colony forming units (CFUs) [42]. At a ratio of 0.5:1, the complex was as potent as the drug alone and was also shown to release amikacin upon addition of acid as the protonated pillar[5]arene precipitates. Amikacin is active against both Gram-positive and Gram-negative bacteria, but is required at high concentrations to be of therapeutic use. Therapeutic concentrations can cause side-effects including nephrotoxicity and ototoxicity, so a method of targeting delivery such as pH-activated release would improve its therapeutic profile [43].

A more complex pillar[5]arene, **65**, was prepared by He and colleagues to deliver vancomycin to MRSA within cells (Figure 12) [44]. The pillar[5]arene was decasubstituted with substituents incorporating a hydrophobic region attached by click chemistry to an acid-sensitive lysine-phenylalanine region, which was linked in turn by thiourea to a terminal mannose. Upon addition of vancomycin, self-assembly resulted in the formation of vesicles with externally facing mannose groups able to bind to mannose receptors prior to endocytosis. The pillar[5]arene-vancomycin vesicles were found not to affect bacterial growth, but once inside cells, could be degraded by cathepsin A or low pH to release vancomycin. Pillar[5]arene vesicles formed in the presence of the fluorescent dye propidium iodide were used to demonstrate that vesicle release occurred within macrophages. MTT assays on three cell lines found no cytotoxic effects. To assess the ability of vancomycin-containing vesicles to target intracellular MRSA, RAW264.7 cells, originally from leukaemia-transformed murine cells, were infected with the WHO-2 strain of MRSA. Vancomycin released into cells was measured by HPLC and was consistent with the incubating concentrations of the drug. The impact on cells was determined by observing the number of CFUs from initial administration of the vancomycin-containing vesicles, which revealed a decrease of two orders of magnitude over 25 h.

A summary of the antibiotic activities of the most active macrocyclic derivatives is given in Table 1.

## 3. Cell Destruction

As an alternative mode of action to drug delivery or release of toxic metal ions, the amphiphilic properties of macrocycles can be utilized to disrupt bacterial and fungal membranes, leading to rupture and cell death. Where the compounds are charged, there may be an even greater affinity for biological membranes and, potentially, different responses to Gram-positive and Gram-negative bacteria, and to fungi, which have a different membrane composition.

### 3.1. Macrocyclon

In 1951, Cornforth and colleagues reported on the suppressive effects of the non-ionic surfactant Triton A20 on tuberculosis in mice [45]. The surfactant consists of a 4-(2,4,4-trimethylpentan-2-yl)phenol head group, or ‘octylphenol’, linked to polyether substituents of varying lengths formed through reaction with ethylene oxide. In a later paper, reaction of these compounds with their 2,6-bis(hydroxymethyl) analogues resulted in complex mixtures from which linear polymers comprising an odd number of octylphenyl units could be isolated [5]. No cyclic products were found, so the group used the Zinke method to prepare a macrocycle from octylphenol and formaldehyde [46]. The condensation product was believed to be a cyclotetramer, or *cyclo*tetra-*m*-benzylene, and subsequently reacted with ethylene oxide to give *Macrocyclon*, **66**, a macrocyclic analogue of the authors’ linear polymer shown in Figure 1 and Figure 13. It was essentially non-toxic to mammals and, unlike its linear analogue, found to be more potent than streptomycin in the treatment of the human virulent strain of *M. tuberculosis*, H37Rv. The greatest effects were seen where the polyether comprised 15 to 20 repeat units, but when this extended to 45 units, the compound became pro-tuberculous. None of the derivatives inhibited growth of tubercular bacilli and it was suggested that the activity was due to their surfactant effects. In a later paper, D’Arcy Hart and colleagues revisited these compounds and compared *Macrocyclon*, with an average of 12.5 -CH_2_CH_2_O- repeat units and terminating in a hydroxy group, with ‘HOC-60’, **67**, an analogue with 60 repeat units [47]. At this point, it was also appreciated that the compounds were calix[8]- rather than calix[4]arene derivatives. It was observed that growth of *M. tuberculosis* inside macrophages was inhibited by *Macrocyclon*, but stimulated by HOC-60. Similarly, lipase activity was inhibited by *Macrocyclon* and stimulated by HOC-60, suggesting that lipids and lipid metabolism were affected by the macrocycles.

Almost 50 years after the original report, *Macrocyclon* was reinvestigated by Tascon and colleagues, who confirmed the original findings [48]. In these later experiments, both macrophages and live mice were infected with *M. tuberculosis* and the range of calixarenes, **68** to **72** (Figure 13), increased.

Treatment with *Macrocyclon* supported the novel therapeutic pathway proposed by D’Arcy Hart, as the macrocycle enhanced the innate defense mechanisms in the murine macrophages. This work was followed up by a much more extensive screen of 25 calixarenes, **41**, **66**, and **68** to **91**, against *M. tuberculosis* by Hailes and colleagues (Figure 14) [49]. Derivatives based on calix[4], -[6], -[7], and -[8]arenes with *t*-butyl, phenyl, and sulfonate upper rim substituents, and a range of lower rim, largely ethylene glycol substituents, were assessed alongside *Macrocyclon*. The parent 4-*t*-butylcalix[8]arene and 4-phenylcalix[8]arene were more active than their smaller homologues and the 4-sulfonatocalix[8]arene, being water soluble, had activity approaching that of *Macrocyclon*. The addition of polyethylene glycol substituents enhanced the calixarenes’ anti-mycobacterial properties, with complete substitution having greater effects than partial substitution. Longer substituents were required for 4-phenylcalix[7]arene derivatives to be effective, presumably owing to the parent compound’s lower solubility compared with the 4-*t*-butylcalixarenes. Lower rim acetate groups elicited pro-tubercular activity and other substituents, such as cyanopropoxy groups, had little effect.

### 3.2. Charged Calixarenes

Regnouf-de-Vains and colleagues also investigated the anti-mycobacterial activities of 17 charged calix[4]arenes [50,51,52]. In addition to parent compounds **5**, **32** to **34**, and **92**, derivatives with bithiazolyl lower rim subunits, **26**, **29**, and **31**, previously tested as antibiotics [28], and bipyridyl analogues with sulfonate, carboxylate, phosphonate, ethylamine, or ethylguanidinium upper rim termini, **92** to **104** (Figure 15), were also assessed for activity against *M. tuberculosis* H37Rv. None of the anionic species exhibited activity, but the parent 4-guanidinoethylcalix[4]arene, **5**, and its derivative with two lower rim bipyridyl substituents, **96**, provided almost 100% inhibition at concentrations of 1.22 μg mL^−1^ and 1.89 μg mL^−1^, respectively.

Independently, the groups of Yushchenko [53] and Loftsson [54] focused on cationic calix[4]arenes with trimethylammonium, *N*-(2-hydroxyethyl)-*N*,*N*-dimethylammonium, or *N*-(2-aminoethyl-*N*,*N*-dimethylammonium groups linked to the upper rims by methylene groups, and O-methyl, O-propyl, or O-octyl lower rim substituents, **105** to **108** (Figure 16). Derivatives **105** and **106** showed increasing hemolysis from 0.71 μg mL^−1^ and 0.89 μg mL^−1^, respectively. The antimicrobial activity of 2-hydroxyethyl derivatives **105** and **108**, together with N-methylimidazolium derivatives **109** and **110** (Figure 16) against *S. aureus* 209 P and *E. coli* M-17 bacteria, was studied. Compounds **105** and **109** had pronounced activity with MICs of 1.95 μg mL^−1^ for *S. aureus* 209 P, while **107** and **108** had no such activity and all compounds were inactive against the Gram-negative *E. coli* M-17. The authors concluded that the antibacterial activity depended on the size and conformational rigidity of the macrocycle and the length of the alkyl substituents on the lower rim of the calixarene. Conformationally rigid macrocycles **106** and **109** were active, while conformationally flexible **108** was inactive. In addition, with an increase in the length of the lower rim alkyl substituents from propyl to octyl, the antibacterial activity decreases. Tests on *S. aureus* ATCC 29213 and *E. coli* ATCC 25922 and M-17 showed that **106** and **107** had very similar profiles to the guanidinoethylcalix[4]arenes reported by Regnouf-de-Vains.

Melezhyk and colleagues also investigated **106** and **107** together with the four cationic derivatives, **109** to **112** (Figure 16) [55]. Tests against *P. aeruginosa* ATCC 9027, *S. aureus* ATCC 6538, *K. pneumonia* ATCC 10031, and *E. coli* ATCC 25922 found that **105**, **109**, and **110** had MICs of 10 μM against *S. aureus* ATCC 6538, but the remaining calixarenes were ineffective. Other than **109**, with an MIC of 1.11 μg mL^−1^, none of the compounds were effective against *K. pneumonia* ATCC 10031 or *E. coli* ATCC 25922, and only **111** had biofilm inhibiting activity, but this was seen for all strains except *P. aeruginosa* ATCC 9027.

Consoli and colleagues designed calixarene, **113**, which binds a nitric oxide (NO)-releasing guest in response to irradiation at 400 nm (Figure 17) [56]. Switching between ‘dark’ and ‘light’ conditions allowed NO release to be followed electrochemically. The calixarene itself had no observable antimicrobial activity against either *S. aureus* ATCC 6538 or *P. aeruginosa* ATCC 9027, but when combined with the otherwise insoluble guest, approached 100% inhibition for both after 20 min of illumination. The group developed a more soluble derivative of the calixarene, **114** (Figure 17), which incorporated aspects of the NO-releasing species to determine if this would impart antibiotic activity [57]. Experiments on human skin fibroblasts showed negligible antiproliferative activity, however, disc diffusion experiments and observation of CFUs demonstrated a 98.9% reduction of *S. aureus* ATCC 6538 CFUs in the dark and a 99.95% reduction after 20 min irradiation. *E. coli* ATCC 10536 was unaffected in the dark, but after 30 min irradiation, it saw a 93.5% reduction of CFUs. The difference was ascribed to the combination of broad bactericidal NO release, which affected both *S. aureus* and *E. coli*, and the quaternary ammonium groups in the calixarene framework, which had a greater disruptive effect on the cell membranes of Gram-positive *S. aureus*.

Di Fatima and colleagues prepared a family of iminocalixarenes, **115** to **120** (Figure 17), and investigated their antifungal properties compared with their monomeric analogues [58]. MIC tests against *C. albicans* ATCC 18804, *Candida krusei* ATCC 20298, *Candida tropicalis* ATCC 750, *Candida parapsilosis* ATCC 22019, *Candida glabrata* ATCC 90030, and *Candida dubliniensis* CBS 7987 showed the calixarenes to have up to a 36-fold enhancement active against *Candida* strains compared with the monomers. The antifungal response of **115** was generally comparable to that of fluconazole and twice as active against *C. krusei*.

### 3.3. Vancomycin Mimicking Calixarenes

A different approach to imparting antibiotic activity to calixarenes was pioneered by Ungaro and colleagues in 1996 [59]. Vancomycin mimics were prepared by bridging two opposite rings in *O*-propyl calix[4]arene through their upper rims with two, **121**, or four alanine residues, **122**, linked by diethylenetriamine, as shown in Figure 17. The compounds’ effects on three strains of *S. aureus* (penicillin-sensitive 663, penicillin-resistant 853, and methicillin-resistant 1131) together with *Bacillus cereus*, *Saccharomyces cerevisiae*, *Acholeplasma laidlawii*, *S. epidermidis*, *E. coli* 1852, and *C. albicans* were determined and compared to that of vancomycin. Significant activity against *S. epidermidis* and all strains of *S. aureus*, with MICs between 4 μg mL^−1^ and 16 μg mL^−1^, were found for the -D-Ala-NHCH_2_CH_2_NHCH_2_CH_2_-D-Ala- and its L-Ala analogue. These compared favorably with 2 μg mL^−1^ seen for vancomycin. Neither vancomycin nor the alanine-bridged calixarenes showed activity against *S. cerevisiae*, *A. laidlawii*, or *C. albicans*.

### 3.4. Other Calixarenes

Jain and colleagues prepared a number of oxacalix[4]arenes, including **123** and **124** (Figure 18), which exhibited a range of activities against bacterial and fungal strains [60]. Compound **123** was particularly effective against *E. coli*, *P. aeruginosa*, and *S. aureus*, with MICs below 2 μg mL^−1^, whereas **124** was additionally effective against the fungal strains *C. albicans* and *Aspergillus clavatus*, with MICs of 12.5 μg mL^−1^ in both cases.

In 2012, Memon and colleagues tested the antibacterial and fungicidal activity of a calixarene with morpholine groups introduced to the upper rim, **125** (Figure 19), based on the antifungal properties of morpholine derivatives [61]. Using the disc diffusion method, *Staphylococcus albus* ATCC 10231, *Streptococcus viridans* ATCC 12392, *Bacillus procynous* ATCC 51189, *Enterobacter aerogenes* ATCC 13048, *Klebsiella aerogenous* ATCC 10031, *E. coli* ATCC 8739 *Sallmonella* ATCC 6017, *A. niger* ATCC 16404 *Aspergillus fumagatus* ATCC 90906, and *Penicillium* ATCC 32333 were used as a range of organisms that covered Gram-positive, Gram-negative, and fungal strains. Calixarene **125** exhibited excellent activity against all strains except *S. albus*, with MICs between 4 and 8 mg mL^−1^. Continuing with this theme, the group synthesized the compound’s pyrrolidine analogue, **126** (Figure 19) [62]. The MIC values were in the range from 1170 to 2340 μg mL^−1^ for *S. aureus*, *S. viridans*, and *E. coli*, and from 580 to 2340 μg mL^−1^ for fungal strains *A. niger*, *A. flavus*, and *C. albicans*. The group also investigated the effect of 4-nitrocalix[4]arene, **127** (Figure 19), on a similar range of strains [63]. Disc diffusion experiments using *S. aureus*, *S. viridans*, *E. coli*, *A. niger*, *Aspergillus flavus,* and *C. albicans* showed good antibacterial activity against *S. aureus* and *S. viridans*, with MICs below 5000 μg mL^−1^, together with *A. flavus* and *C. albicans*, at 2300 μg mL^−1^, but the best results were against *E. coli* and the fungal strain *A. niger*, with an MIC of 580 μg mL^−1^ for both. In support of their findings, the authors state that phenol-containing compounds with electron-withdrawing groups in the 4-position increase activity.

Morpholine and pyrrolidine groups were chosen as lower rim substituents by Galitskaya and colleagues [64]. These two 4-*t*-butylthiacalix[4]arene derivatives, **128** and **129** (Figure 19), formed nanocomposites with silver cations that were used to treat polyethylene films. The *cone* conformer was found to be the most active, totally inhibiting the growth of *P. aeruginosa* on the film’s surface after 24 h; however, no effects were seen for *Bacillus pumilus*.

### 3.5. Pore-Forming Pillar[5]arenes

Pillar[5]arenes can be modified to incorporate short amino acid substituents that allow them to penetrate phospholipid membranes. Hou and colleagues prepared pillar[5]arenes **130** to **133** (Figure 20) with substituents comprising one to four amino acids (-L-Trp-CO_2_H, -D-Leu-L-Trp-CO_2_H, -L-Leu-D-Leu-L-Trp-CO_2_H and -D-Leu-L-Leu-D-Leu-L-Trp-CO_2_H) based on the terminus of gramicidin A, a natural pore-forming protein [65]. Tests on *E. coli*, *S. epidermidis*, *S. aureus*, and *B. subtilis* showed significant growth inhibition by all compounds at concentrations of 10 μM (from 3.05 μg mL^−1^ for **130** up to 6.64 μg mL^−1^ for **133**) on the Gram-positive bacteria, but had no effect on *E. coli*.

Xin, Dong, Chen, and colleagues undertook similar experiments with five pillar[5]arenes, **134** to **138** (Figure 21), with two helical peptides joined to the same ring [66]. Here, the peptides were between 8 and 16 residues long and also tested against *E. coli*, *S. epidermidis*, *S. aureus*, and *B. subtilis*. An interesting correlation was observed between activity and length of peptide chain for the Gram-positive bacteria and the 16-residue derivative approached the IC_50_ of gramicidin A. Much lower activity was seen against *E. coli* with 30% of bacteria surviving even at concentrations of 500 μM (equating to 138 μg mL^−1^ for **134** up to 235 μg mL^−1^ for **135**).

A summary of the cytotoxic activities of the most active macrocyclic derivatives is given in Table 2.

## 4. Biofilm Inhibition

Direct delivery of a drug or prodrug to individual cells is one antimicrobial strategy, however, disruption of bacterial colonies growing on a surface is another approach. Once individual bacteria are able to bind to a surface, initially through hydrophobic and other weak interactions, they can act as anchors for further cell deposition. Biofilms then form when the bacteria are able to deposit an extracellular matrix of polymeric material, largely composed of polysaccharides, which allows them to extend the colony [67]. The polysaccharides are recognized by lectins, carbohydrate-binding proteins present on cell surfaces, leading to the formation of a cellular matrix. These complex, surface-bound films accumulate more bacteria, facilitating chemical communication between them. One consequence of this is increased resistance to antibiotics and surfactant-based detergents as the outer layers form a protective barrier for the remainder of the colony. While the biofilm may comprise a single species, it is possible for synergistic ecosystems to emerge in which molecules generated by one species are metabolized by others. As different species of bacteria have different susceptibilities to antibiotics, biofilm heterogeneity can also confer a level of overall protection.

While antibiotics and surfactants are widely used against biofilms, inhibition of their formation and growth has been approached using multivalent macrocycles. Several groups have developed derivatives with extended substituents terminating in saccharides such as mannose [68,69]. The sugar termini bind to receptor sites on bacterial proteins. These sites would usually interact with polysaccharide groups in the extracellular matrix, so blocking them inhibits adhesion. Calixarenes, along with related macrocycles such as pillararenes, incorporating several substituents have been developed as inhibitors. A single macrocycle with several sugar groups has two benefits. Its multivalency maximizes the chances that it will block binding sites and extended substituents are able to span between several receptor sites on the same protein. In doing so, they can form a physical barrier between the protein surface and the extracellular matrix. Consequently, the conformation of the calixarene is often a vital factor in the compound’s effectiveness. The *cone* conformer of calix[4]arene is able to bind four sites on the same protein face and is essentially tetravalent, whereas the same compound in the *1,3-alt* conformer would have a greater span between binding sites, but would only be divalent.

Lectin binding by macrocyclic glycoconjugates can also reduce infection by stopping the cell adhesion process from the very beginning. With carbohydrate recognition sites blocked, bacteria have to rely on weak protein–surface interactions to aggregate and these are easily reversed. To determine the efficacy of the macrocyclic agents, two key factors need to be quantified: affinity and inhibition. Experiments undertaken in the presence of different lectins will reveal specificity and affinity, while biofilm inhibition properties are evaluated through cell culture experiments.

### 4.1. Calixsugars

In 2009, Imberty, Matthews, Vidal, and colleagues prepared a family of 4-*t*-butylcalix[4]arenes that were partially *O*-alkylated with propyl bromide and then *O*-propargylated at the remaining lower rim positions [70]. Acetylated mannose or galactose moieties were attached by a triethylene glycol tether using click chemistry. Galactose derivatives with one, two, or three substituents were isolated and deprotected to give **139** to **142** (Figure 22). The fully substituted derivatives were synthesized from the tetra-*O*-propargyl precursor, which gave rise to *cone*, *1,3-alt*, and *partial cone* products **143** to **145** (Figure 22). The tetrasubstituted mannose analogue, **146** (Figure 22), was also prepared, in the *cone* conformer, as a comparator. Isothermal titration calorimetry was used to show binding to Lec A (galactophilic *P. aeruginosa* first lectin, or PA-IL) and that the tetravalent galactose ligands bound most strongly. Enhanced binding of the *partial cone* and *1,3-alt* conformers demonstrated the importance of geometric alignment when binding to the protein surface. Surface plasmon resonance (SPR), and later atomic force microscopy (AFM) and computer modelling studies [71], supported these findings. Imberty, Vidal, and colleagues then investigated the influence of the linker arm between the calixarene and sugar terminus [72]. Linkers with increasing rigidity, but similar lengths, shown in Figure 22 and incorporating ethylene glycol, **147**; diethylene glycol acetamide, **148**; ethylene glycol acetamidoacetamide, **149** and **150**; and phenyl acetamidoamide, **151** to **153**, were compared with the original triethylene glycol linker. Binding to PA-IL was assessed by SPR, isothermal titration microcalorimetry (ITC), hemagglutination (HIA), and enzyme-linked lectin assays (ELLA). The cost of increasing rigidity was lower solubility, nevertheless, the researchers found that the calixarene with four diethylene glycol acetamide linkers was more potent than the original trethylene glycol-containing compounds, but only when in the *1,3-alt* conformation.

Using a similar, but simpler approach, Consoli, Geraci, and colleagues synthesized a tetra-*O*-propyl *cone*-calix[4]arene, **154** (Figure 23), with fucose groups on the upper rim by an acetamidoacetamide linker [73]. A planktonic antimicrobial susceptibility test was performed with wild-type *P. aeruginosa* PAO1 using the microdilution broth method and no antimicrobial activity was seen up to 32.7 μg mL^−1^. Biofilm inhibition was observed, however, with a dose-dependent response seen over the range from 2.0 μg mL^−1^ to 32.7 μg mL^−1^, with concomitant inhibition rates rising from 35% to 73%. The precursor calixarene without the sugar groups, terminating in four amines, also actively inhibited biofilm formation, but to a lesser extent. The authors ascribe this to interactions between the bacteria and the formation of positively charged ammonium groups.

Returning to the triethylene glycol spacer motif, Vidal and colleagues extended the family of calixarenes to include those with four fucose termini, **155** (Figure 23) [74]. With glucose, galactose, mannose, and fucose analogues in hand, their effects on bacterial aggregation, cell adhesion, and biofilm inhibition could be compared. As LecA is selective for galactose and LecB (fucophilic *P. aeruginosa* second lectin, or PA-IIL), discrimination between the two should prove possible. ITC was used to determine affinities of the compounds for LecA and LecB. A monovalent fucosylated linker, prepared as a control, bound to LecA with an affinity in the nanomolar range, in agreement with similar literature examples. The data suggested that a *1,3-alt* calixarene with four fucose-terminated substituents appeared to utilise three of them when binding and, as expected, its affinity for LecB was 50 times lower. The glucosylated analogue was not recognised by either lectin. *P. aeruginosa* wild-type PAO1 strains and those overproducing either LecA or LecB were used in aggregation experiments. Little aggregation was observed with galactosylated of fucosylated calixarenes for wild-type *P. aeruginosa*, but the results from PAO1∆*lecA* and PAO1∆*lecB* implied that clustering was LecA-dependent and LecB-independent. Cell adhesion experiments using A549 lung epithelial cells showed dose-dependent inhibition of between 70% and 90%, regardless of which calixarene was tested, but this was significantly better than the linkers alone. Inhibition of LecA- and LecB-dependent biofilms after 24 h was observed at calixarene concentrations of 1.1 μg mL^−1^ or higher, regardless of the terminal sugar, but did not affect bacterial growth.

In later work, Geraci and colleagues altered the structure of the upper rim slightly to give a more extended reach for the calixarene’s fucose termini, **156** (Figure 23) [75]. Biofilm inhibition tests with *P. aeruginosa* (PAO1) and *S. epidermidis* ATCC 35984 showed 35.4% inhibition of PAO1 at 27.9 μg mL^−1^ and complete inhibition between 55.7 μg mL^−1^ and 111.5 μg mL^−1^ with similar results for *S. epidermidis*. Monomeric analogues had some effect against PAO 1, but enhanced *S. epidermis* biofilm formation. The authors note that their results for a fucose-terminated calixarene in the *cone* conformation were orders of magnitude better than those obtained by Vidal for **155** in the *1,3-alt* conformation [64].

Benazza and colleagues prepared fuco-, **157**; manno-, **158**; and glucocluster, **159**, *cone*-calix[4]arene derivatives, shown in Figure 23 and Figure 24, and introduced an iron chelating hydroxamic acid region in each linker [76]. It was proposed that, in addition to binding to lectins, these compounds could sequester soluble iron, an essential nutrient for bacteria, to enhance their antibacterial activities. No antibiotic activity against wild-type *P. aeruginosa* PAO1 was seen for the fucose or mannose terminated derivatives, so additional experiments investigated the effects of iron chelation. Under iron-depleted conditions, production of the bacteria’s main siderophore, pyoverdine-I (Pvd-I), increased to counteract the sequestering effects of the calixarenes. When a siderophore-deficient strain of *P. aeruginosa* was used, the mannosylated calixarene significantly disrupted growth. Binding assays using fluorescence polarization revealed an unexpected interaction between LecB and the mannosylated calixarene over 200 times stronger than a monomeric analogue. Computer modelling indicated that the calixarene could bind simultaneously to four sites on the LecB protein, which may explain this effect. *P. aeruginosa* PAO1 biofilm growth studies at 5.4 μg mL^−1^ found that not only did the mannosylated-derivative inhibit biofilm formation by 84%, but the fucoslyated-derivative did so by 72%. Even more surprising was the effect of the glucosylated-derivative, used as a negative control, which gave 92% inhibition. The authors concluded that other inhibitory mechanisms could be involved such as the possibility that nitric oxide was being released from the iron-binding site.

### 4.2. Biofilm Inhibiting Resorcinarenes

A much simpler approach to biofilm inhibition was taken by Guildford and colleagues, who assessed the effects of a polyethylene glycol-bound resorcinarene catheter coating on biofilms of *E. coli* NCTC 10418 and *Proteus mirabilis* NCTC 11938 [77]. Although the macrocycle’s solubility is often an issue, this can be improved through functionalisation with polyethylene glycol, **160** (Figure 25). The incorporation of upper rim silane groups facilitates its attachment to silicone coated devices [78]. The study found that the coating was able to significantly increase biofilm inhibition of *P. mirabilis* over 10 days, but was less effective against *E. coli*. Resorcin[4]arenes had also been used by Vidal, Matthews, and colleagues as tetravalent ligands for LecA and the lactose-specific lectin, galectin-1 (Gal-1) [79]. Resorcin[4]arene derivatives terminating in either galactose or lactose groups were synthesized in both chair (*rctt*) and boat (*rccc*) stereoisomers, **161** to **164**, as shown in Figure 25. Solubility issues resulted in only certain combinations of compounds and lectins being assessed for binding. Compared with a monomeric analogue, the galactose-terminated resorcin[4]arenes had a lower IC_50_ by a factor of 240 to 300, but no significant differences arose as a result of their topology. Molecular modelling suggested that two resorcin[4]arene substituents bound to adjacent LecA monomers in a similar manner to calix[4]arene-based analogues.

Calix[4]resorcinarenes were also used by Kashapov and colleagues [80]. Two derivatives, **165** and **166** (Figure 25), incorporating *N*-methyl-D-glucosamine at the upper rim were prepared and assessed for their activities against *S. aureus* 209 P, *Bacillus cereus* 8035, *P. aeruginosa* 9027, *E. coli* F-50, *Trichophyton* mentagrophytes var. gypseum 1773, *Aspergillus niger* BKMF-1119, and *C. albicans* 885-653. Both compounds were selective for *S. aureus* 209 P, and **166** had activity against *Bacillus cereus* 8035, with MICs ranging from 17.2 μg mL^−1^ to 137.2 μg mL^−1^. The authors suggest that both compounds can form aggregates, which was supported by NMR diffusion spectroscopy, but that differences in their structures occur due to the more hydrophilic nature of **166**.

Several other groups have also employed resorcinarenes. Utomo and colleagues reported on a C-4-methoxyphenyl[4]resorcinarene with hexadecyltrimethylammonium substituents, **167** (Figure 26), which had a higher activity than the unsubstituted parent compound against *S. aureus*, but was ineffective against *E. coli* [81]. Yadav, Kumari, and colleagues used C-methyl[4]resorcinarene, **168** (Figure 26), to complex the antibiotic gatifloxacin and deliver it to *S. aureus* subsp. aureus and the human lung pathogen *Legionella pneumophila* subsp. pneumophila ATCC 33152 [82]. The complex had a lower MIC against *S. aureus* than the antibiotic alone, 0.16 μg mL^−1^ against 0.195 μg mL^−1^, whereas with *L. pneumophila*, an MIC of 0.025 μg mL^−1^ was found whether the drug was complexed or not. Interestingly, the macrocycle itself had no effects on *S. aureus* below 25 μg mL^−1^, but an MIC of 0.78 μg mL^−1^ against *L. pneumophila*. Yamin and colleagues investigated the activity of C-5-bromo-2-hydroxyphenyl-2-methyl-[4]resorcinarene, **169** (Figure 26), against MRSA, *S. aureus*, *E. faecalis*, *Enterobacter aerogenes*, and *P. aeruginosa* [83]. The macrocycle had little effect against *E. aerogenes* and *P. aeruginosa*, but had an MIC of 6.25 mg mL^−1^ against *S. aureus* and *E. faecalis* and an excellent MIC of 1.56 mg mL^−1^ against MRSA.

Not all [4]resorcinarenes have antimicrobial activity. Vagapova and colleagues synthesized aminomethylated [4]resorcinarenes, **170** and **171** (Figure 26), which were found to have no in vitro antimicrobial activity against *S. aureus* ATCC 209 P, *B. cereus* ATCC 8035, *E. coli* CDC F-50, *P. aeruginosa* ATCC 9027, *A. niger* BKMF- 1119, *T. mentagrophytes* 1773, or *C. albicans* 855-653 over the concentration range of 0.97 to 500 μg mL^−1^ [84].

### 4.3. Biofilm Disruption through Drug Delivery

Shah and colleagues used a drug-delivery approach to biofilm disruption by encapsulating the broad spectrum antibiotic clarithromycin within self-assembling nanostructures formed by the amphiphilic *O*-decyl sulfonatocalix[6]arene, **172** (Figure 27) [85]. The nanostructures were in the region of 135 nm in diameter and had a calixarene/drug composition of 5:1, which represented over 50% drug loading efficiency. The drug-filled nanostructures had a 40% lower MIC and IC_50_ against *Streptococcus pneumoniae* ATCC 6303 (antibiotic sensitive strain) than the drug alone, with a similar improvement observed when tested against *S. pneumoniae* ATCC 700669 (antibiotic resistant strain). Biofilm inhibition was similarly improved with minimum biofilm inhibition concentrations for clarithromycin of 19.69 μg mL^−1^ and 44.35 μg mL^−1^ with *S. pneumoniae* ATCC 6303 and ATCC 700669, respectively. When the drug-containing nanostructures were, used these values reduced to 13.69 μg mL^−1^ and 15.97 μg mL^−1^, respectively. Drug-free nanostructures had no effect on antibiotic activity or biofilm inhibition. Efficacy was ascribed to cell penetration by the nanostructures and the inhibition of the normal drug efflux systems due to its encapsulation.

### 4.4. Biofilm-Inhibiting Pillar[n]arenes

The growth in the interest in pillararenes has led to a number of groups to exploit their symmetric, tubular structures and introduce extended substituents. This feature has resulted in their potential as biofilm inhibitors. Imberty, Vidal, and colleagues prepared asymmetric pillar[5]arenes with the intent to galactose and fucose termini, **173** to **175** (Figure 28), connected by two different linkers, on one face and methoxy groups on the other [86]. Cyclisation of the starting monomers 1-(2-bromoethoxy)-4-methoxybenzene or 1-methoxy-4-(2-propynyloxy)benzene led to a mixture of diastereoisomers, which could not be readily separated; thus, to ensure multivalency, symmetric, decasubstituted analogues, **176** and **177** (Figure 28), were prepared as comparators. HIA, ELLA, ITC, and SPR were used to examine the binding properties of the pillararenes with LecA; LecB; and a fucose-selective lectin, BambL, from *Burkholderia ambifaria*. The linker length of the galactose-functionalised pillar[5]arenes was the most important factor. The longer chain derivative displayed a much lower IC_50_, in the micromolar range, and MIC. This was shown to be due to the latter’s ability to interact with five lectin monomers, whereas the former was sterically crowded, owing to short linkers between the sugars and the macrocycle, and it could only interact with three monomers. As expected, the galactosylated derivatives bound LecA better than monomeric analogues and the fucosylated derivatives bound to the fucose-specific lectins LecB and BambL. The asymmetric derivative was specific for BambL, with low nanomolar affinity, as it bound no better to LecB than a monovalent ligand. Although not tested on biofilms, this specificity would undoubtedly impart inhibition where BambL-expressing bacteria were involved. The authors note that, as pillararenes have inherent chirality, both diastereoisomers must be present in a racemic mixture and the effects are an average of the two, but whether or not the chirality is important for recognition is unknown.

Nierengarten and colleagues investigated the effects of increasing the numbers of sugars by incorporating *O*-alkyl substituents with both one and two monosaccharides [87]. Derivatives with different lengths of linkers, **178** to **183** (Figure 29), were also prepared to assess their effects. As expected, the galactosylated derivatives bound to LecA with a longer substituent, giving better results. Binding by the fucosylated derivatives to LecB increased with linker length to a point as the use of two triethylene glycol spacers gave poor results. The derivatives with 20 fucose groups were also superior to their analogues with just 10. Despite its poor result with LecB, the fucose derivative with the longest linkers and 20 sugars gave an IC_50_ value in the picomolar range against BambL, over seven orders of magnitude better than the monovalent control.

In an attempt to incorporate both fucose and galactose termini, the Nierengarten group prepared a number of rotaxanes from pillar[5]arenes, **184** to **187** (Figure 30) [88]. Reaction between the azido-functionalized pillar[5]arene and acetylated fucose or galactose followed by diacylation gave the shuttle and an alkyl thread, terminating in either of those sugars, was formed by the same click methods. The combination of ten galactose units and a difucose thread was highly effective against both LecA and LecB, as it contains complementary sugars to both lectins.

The mannose pillar[5]arene analogue, **188** (Figure 30), had previously been prepared and hemagglutination assays with the pillar[5]arene and an acyclic control showed an almost sevenfold enhancement in inhibition due to the presence of the glycocluster [89]. Later work on the same compound found some inhibition against a bacterial liposaccharide heptosyltransferase, WaaC, involved in lipopolysaccharide biosynthesis [90]. Shortly after Nierengarten’s first report, Huang and colleagues published the synthesis of galactose-appended **189** and asymmetric derivative **190** (Figure 31) [91]. No cellular agglutination was observed for the symmetric pillar[5]arene or an acyclic control, owing to the presence of the mannose groups, but an asymmetric derivative with mannose and *n*-decyl substituents appeared to facilitate bacterial adhesion.

The multivalent potential of the sugar-terminated pillar[5]arenes can be seen in Figure 32. Their fivefold symmetry is shown in the top view of asymmetric pillar[5]arene **173**, with one of its four conformers illustrated, and the effect of extending the polyether linkers to increase the compound’s span can be seen in **176**.

While most of the previous examples use sugar-lectin interactions to inhibit biofilm formation, Cohen and colleagues investigated symmetric pillar[5]arenes, **191** to **196** (Figure 33), with ethyl- and methylphosphonium and ethyl- and methylammonium substituents alongside monomeric analogues [92]. Tests against methicillin-resistant *S. aureus* ATCC 33592 and *E. faecalis* ATCC 29212 showed that the monomers were ineffective, but the pillararenes had excellent inhibiting properties. MBIC_50_ values of the phosphonium derivatives were 0.17 μg mL^−1^ (**191**) and 0.47 μg mL^−1^ (**192**) for both bacteria, as were the ammonium analogues (0.17 μg mL^−1^ for **193** and 0.47 μg mL^−1^ for **194**), indicating that it is the positive charges that are essential for biofilm inhibition. The group expanded their investigation to include imidazolium groups, **197** (Figure 32), and varied the counterions and the size of the pillararene, **198** (Figure 32) [93]. Biofilm inhibitory tests against Gram-positive *S. aureus* subsp. *aureus Rosenbach* ATCC 33592, *S. aureus* ATCC 29213, *S. aureus* BAA/043, *E. faecalis* ATCC 29212, *S. epidermidis* RP62A, and *S. mutans* ATCC 700610 showed pillar[5]- and -[6]arenes with trimethylammonium termini were the most potent, with MBIC_50_ values of 0.20 μg mL^−1^ to 0.79 μg mL^−1^ across all species for pillar[5]arene **195** and 0.11 μg mL^−1^ to 0.79 μg mL^−1^ for pillar[6]arene **198**. By contrast, the negatively charged pillar[5]arene carboxylate derivative **64** had little effect on biofilm formation. Tests of the cationic pillararenes against Gram-negative *E. coli* ATCC 25922 and *P. aeruginosa* PAO1 showed no inhibition.

A summary of the biofilm inhibition demonstrated by the most active macrocyclic derivatives is given in Table 3.

## 5. Conclusions

From Cornforth’s *O*-alkylation of calixarenes with cell membrane-disrupting polyethers, to Ungaro’s vancomycin mimics through to Regnouf-de-Vains’ nalidixic acid-appended calixarene prodrugs, and Matthews’ multivalent lectin-binding calixarenes, calixarenes and their related macrocycles have been demonstrating antimicrobial activity for almost 70 years. During this time, the modes of action have evolved from simple surfactant activity through to drug delivery and, most recently, the use of multivalent macrocycles to bind proteins to block their aggregation. Indeed, it is the chemist’s ability to prepare complex multifunctional derivatives with precise regioisomerism, which is propelling the most recent advances in antimicrobial macrocycles. Control of molecular recognition over extended distances is key in this endeavor and yet it relies on some of the first aspects of calixarene chemistry to be exploited chemically: lower rim substitution and conformer control.

Several themes are beginning to emerge: calixarenes incorporating drug moieties such as penicillin are more effective when coupled to ammonium or guanidinium substituents; metal chelating and releasing calixarenes are not as effective as might be expected; and the ability to use macrocycles as multivalent ligands capable of binding in spatially remote locations has been shown to be a highly effective biofilm-disrupting strategy.

Future development will, no doubt, revolve around substituents with greater specificity for target binding sites, such that the compounds are effective at vanishingly small doses. When those discoveries are made, calixarenes may well become the pharmacist’s magic bullets.

## Figures and Tables

**Figure 1 molecules-25-05145-f001:**
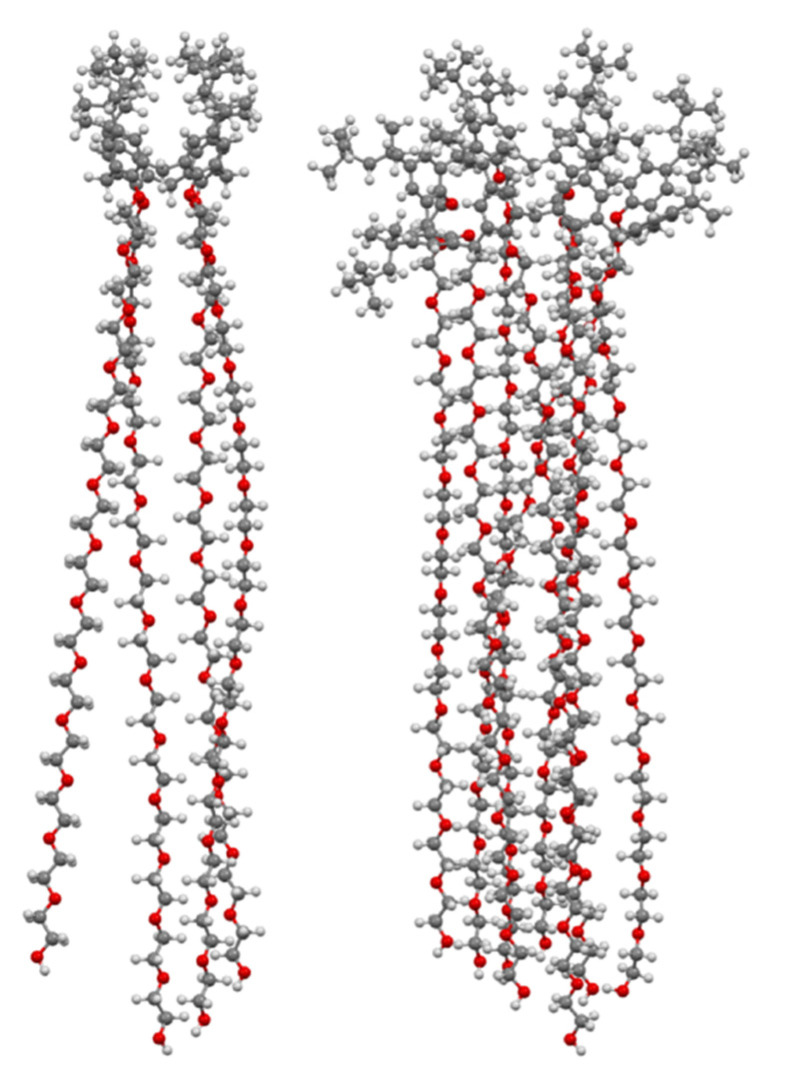
*Macrocyclon* as originally envisaged by Cornforth (**left**) and as later determined (**right**).

**Figure 2 molecules-25-05145-f002:**
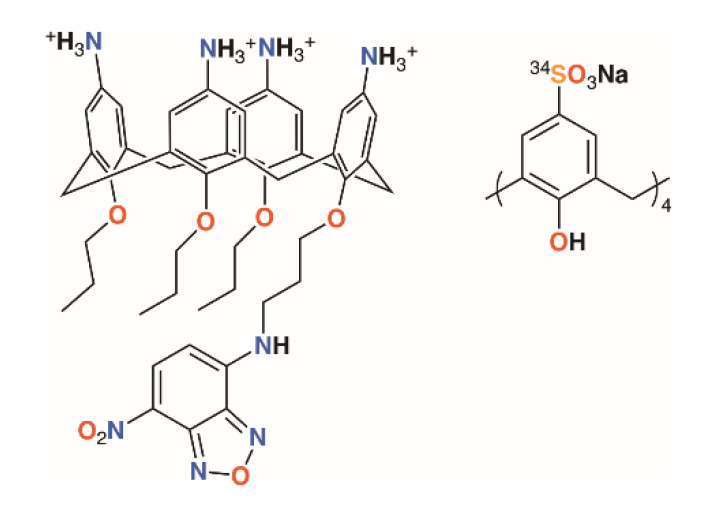
Matthews’ fluorescent calixarene (**left**) and Coleman’s calixarene radiotracer (**right**).

**Figure 3 molecules-25-05145-f003:**
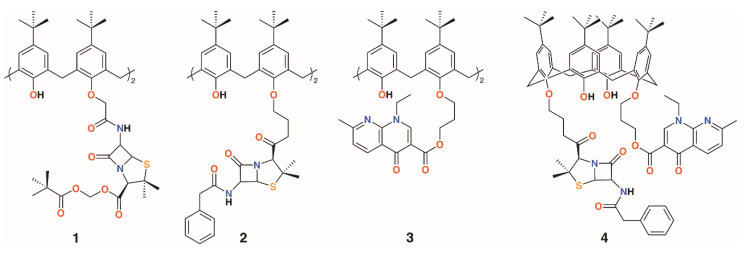
Regnouf-de-Vains’ calixarene prodrugs.

**Figure 4 molecules-25-05145-f004:**
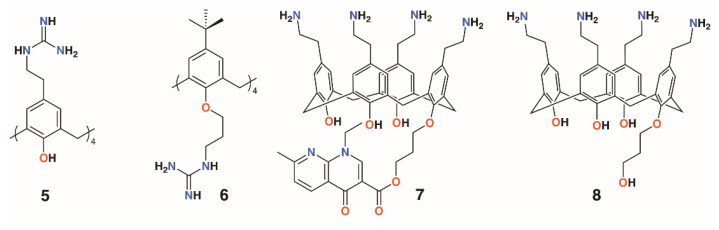
Regnouf-de-Vains’ soluble guanidinium-functionalized calixarenes.

**Figure 5 molecules-25-05145-f005:**
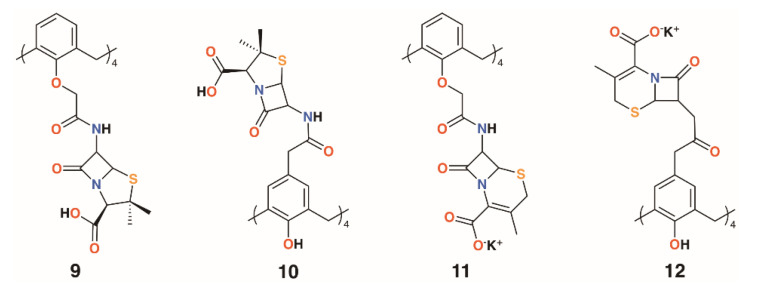
Dillmaghani’s calixpenams and calixcephems.

**Figure 6 molecules-25-05145-f006:**
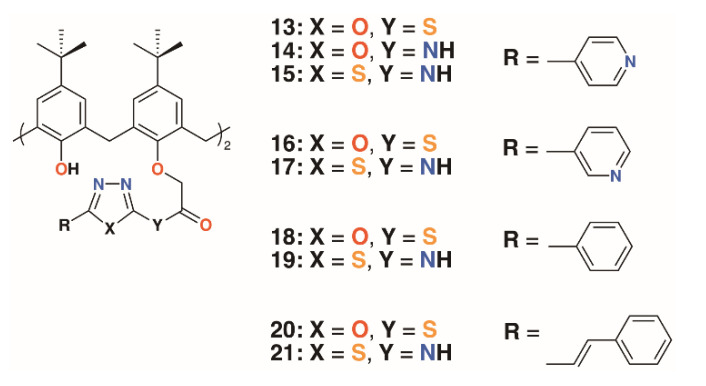
Menon’s’ calixarene prodrugs.

**Figure 7 molecules-25-05145-f007:**
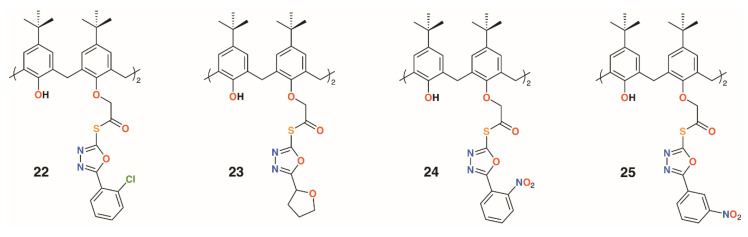
Dillmaghani’s aryl-oxadiazole derivatives.

**Figure 8 molecules-25-05145-f008:**
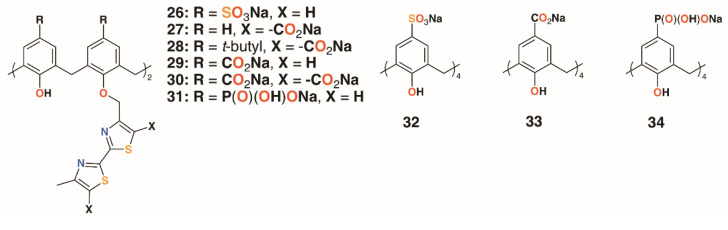
Regnouf-de-Vains’ anti-HIV calixarene prodrugs.

**Figure 9 molecules-25-05145-f009:**
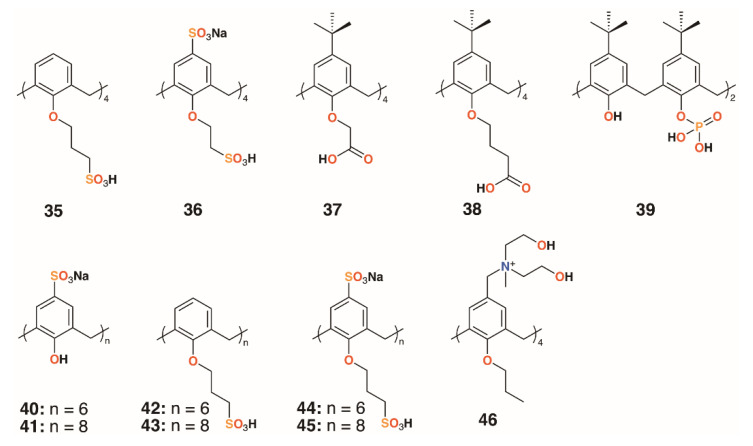
Coleman’s AgNP-capping calixarenes and Nostro’s drug-delivering calixarene.

**Figure 10 molecules-25-05145-f010:**
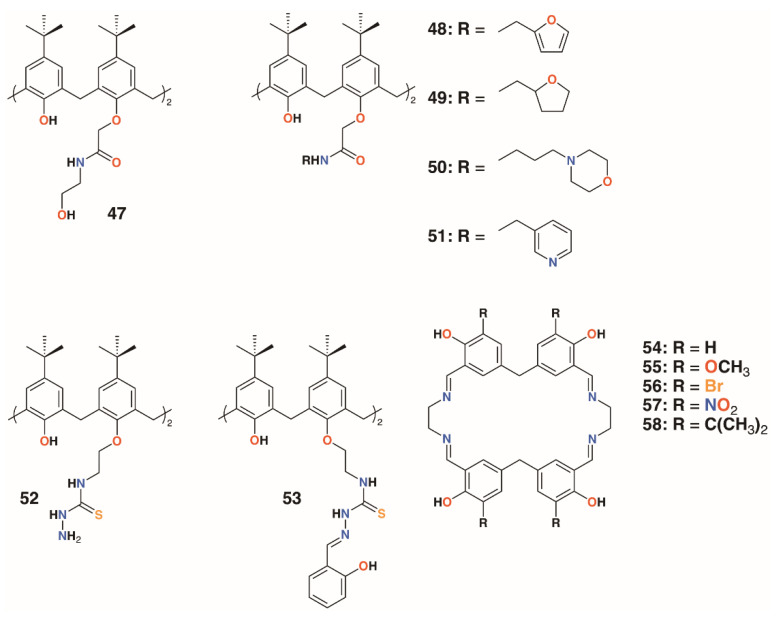
Cation-releasing calixarenes from Memon and Yilmaz with Desai’s corates.

**Figure 11 molecules-25-05145-f011:**
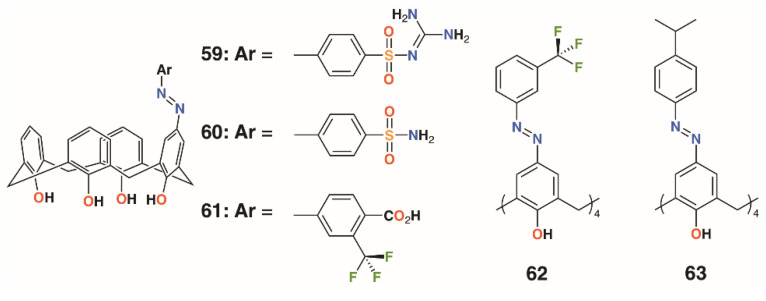
Hamid’s sulfonamides and related compounds.

**Figure 12 molecules-25-05145-f012:**
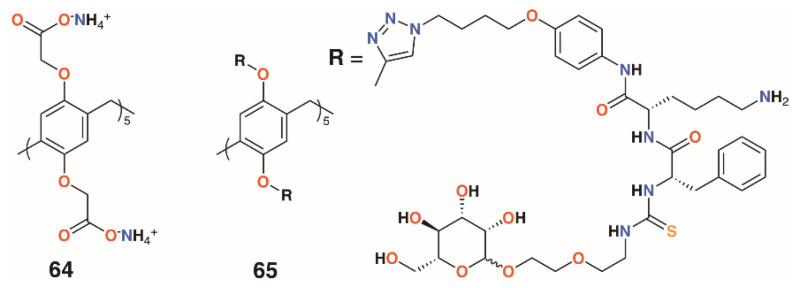
Drug-delivering pillararenes from Notti and He.

**Figure 13 molecules-25-05145-f013:**
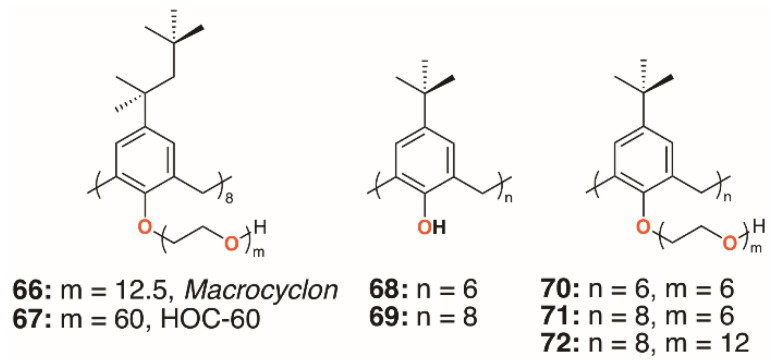
Cornforth’s *Macrocyclon* and HOC-60 with Tascon’s derivatives.

**Figure 14 molecules-25-05145-f014:**
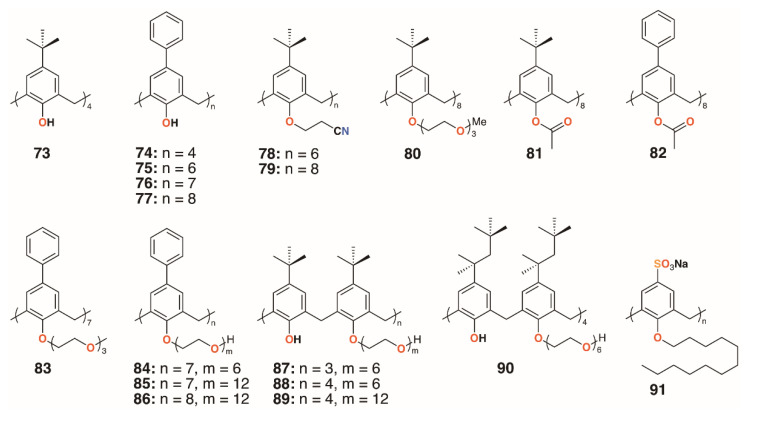
Hailes’ *Macrocyclon* analogues.

**Figure 15 molecules-25-05145-f015:**
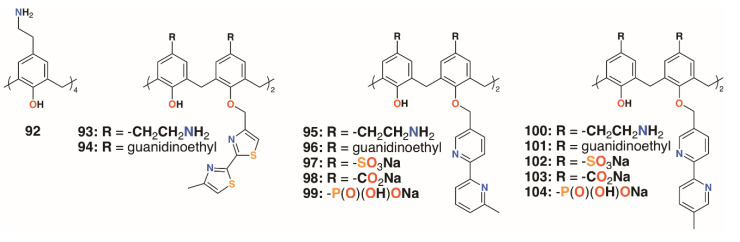
Regnouf-de-Vains’ charged calixarenes.

**Figure 16 molecules-25-05145-f016:**
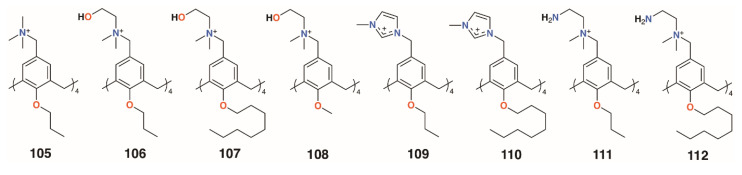
Charged calixarenes reported by Loftsson, Yushchenko, and Melzhyk.

**Figure 17 molecules-25-05145-f017:**
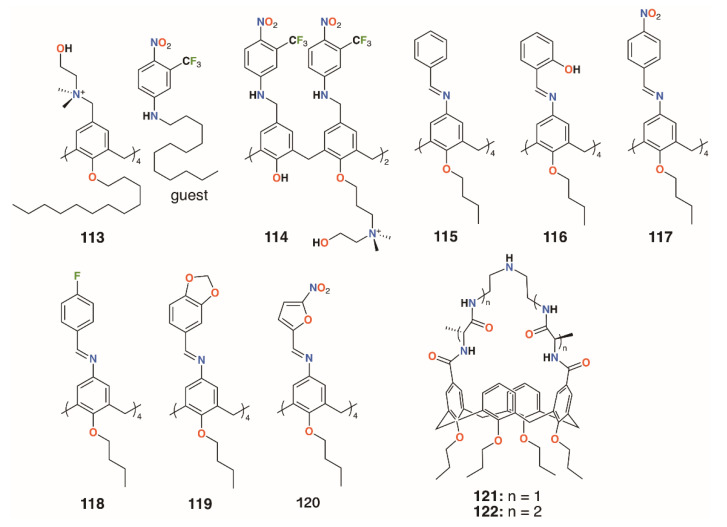
Antibiotic and antifungal calixarenes of Consoli and Di Fatima with Ungaro’s vancomycin mimics.

**Figure 18 molecules-25-05145-f018:**
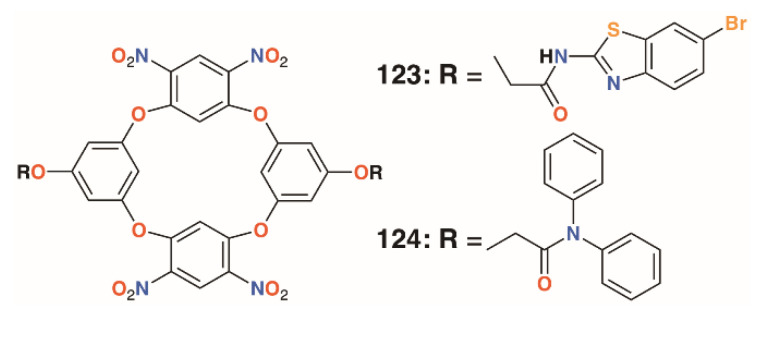
Jain’s oxacalixarenes.

**Figure 19 molecules-25-05145-f019:**
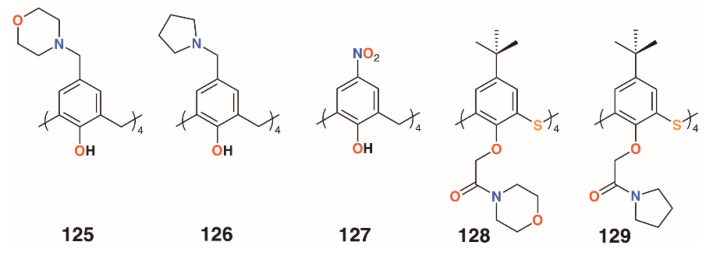
Morpholine and pyrrolidine calixarene derivatives from Memon and Galitskaya.

**Figure 20 molecules-25-05145-f020:**
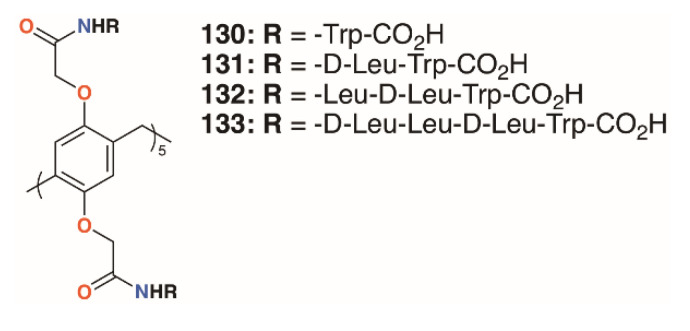
Hou’s pore-forming pillar[5]arenes.

**Figure 21 molecules-25-05145-f021:**
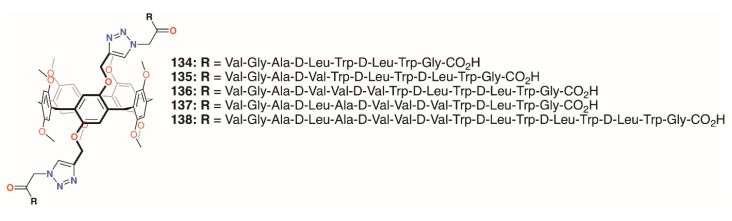
Xin’s pore-forming pillar[5]arenes.

**Figure 22 molecules-25-05145-f022:**
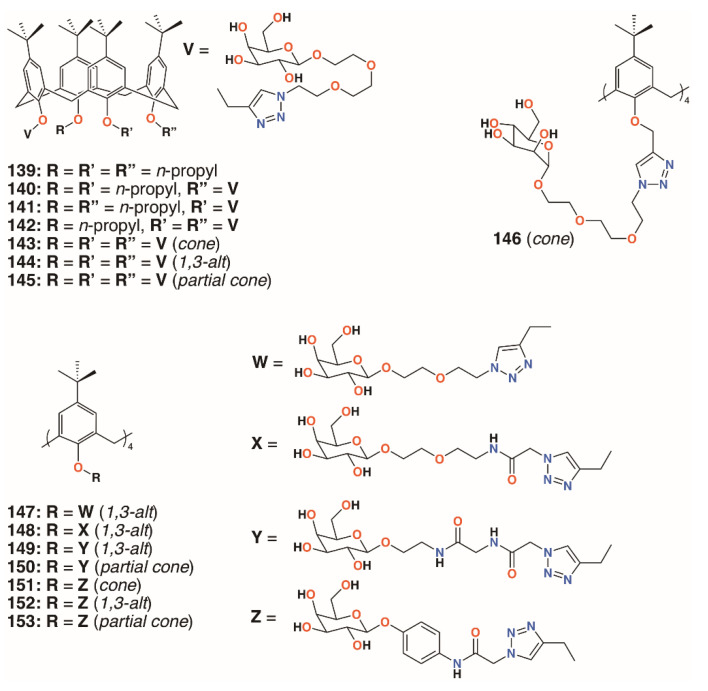
Imberty’s glycosylated calixarenes.

**Figure 23 molecules-25-05145-f023:**
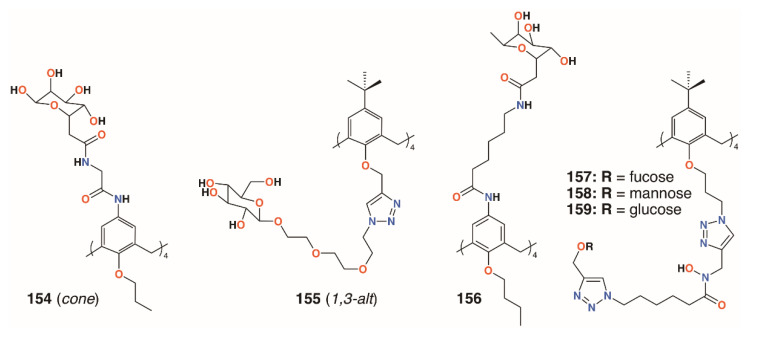
Glycosylated calixarenes from Geraci, Vidal, and Benazza.

**Figure 24 molecules-25-05145-f024:**
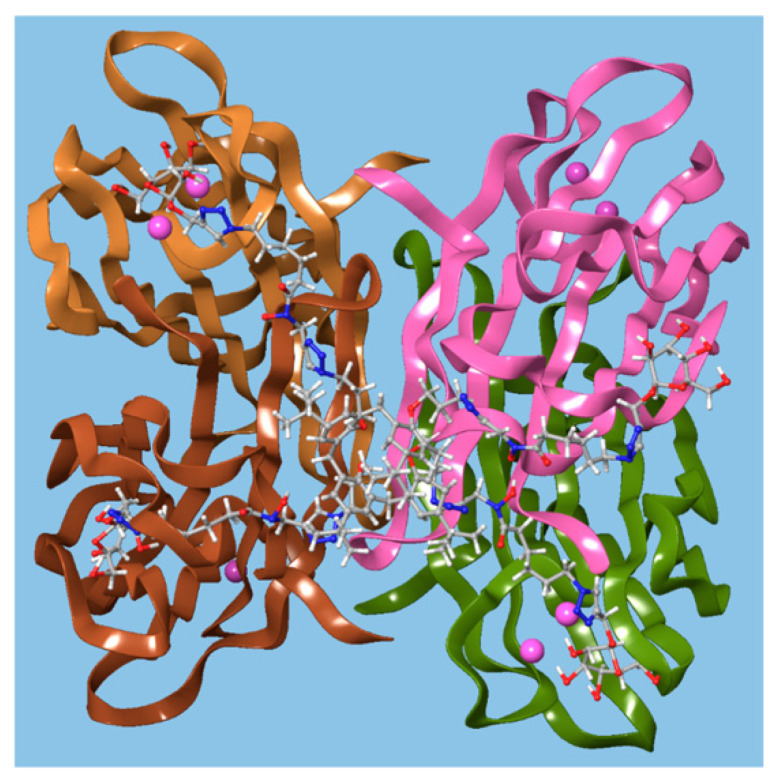
A glycosylated calixarene binding to a lectin. Reprinted with permission from [76]. Copyright © 2019 American Chemical Society.

**Figure 25 molecules-25-05145-f025:**
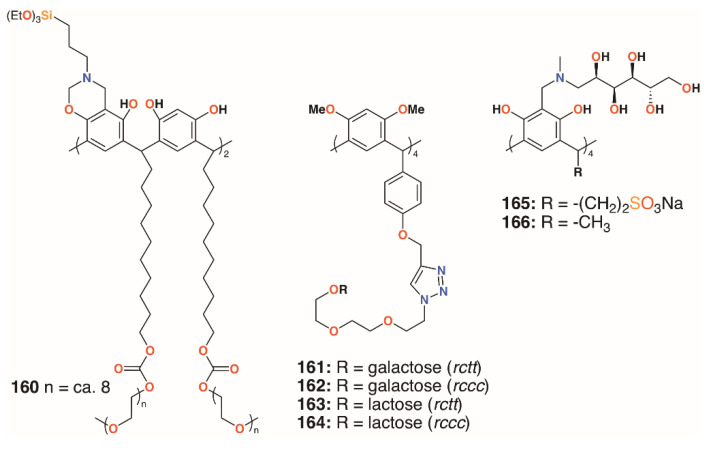
Resorcinarene derivatives of Guildford, Matthews, and Kashapov.

**Figure 26 molecules-25-05145-f026:**
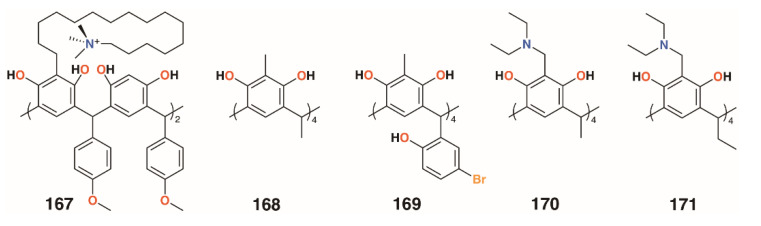
Resorcinarenes from the groups of Utomu, Yadav, Jumari, Yamin, and Vagapova.

**Figure 27 molecules-25-05145-f027:**
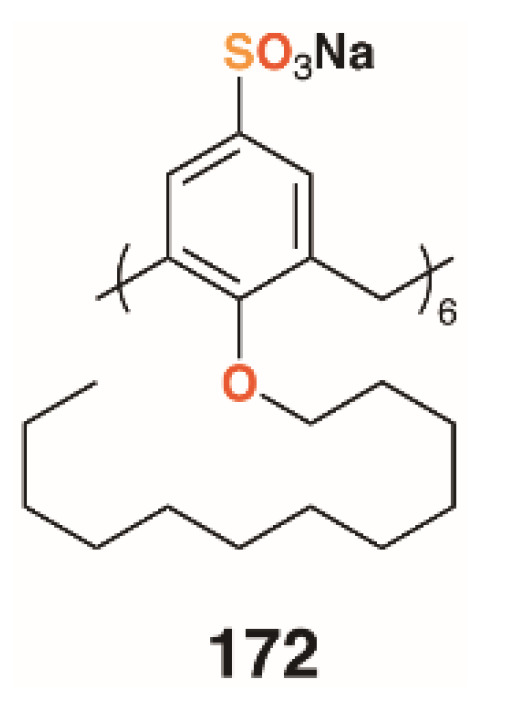
Shah’s nanostructure-forming calixarene.

**Figure 28 molecules-25-05145-f028:**
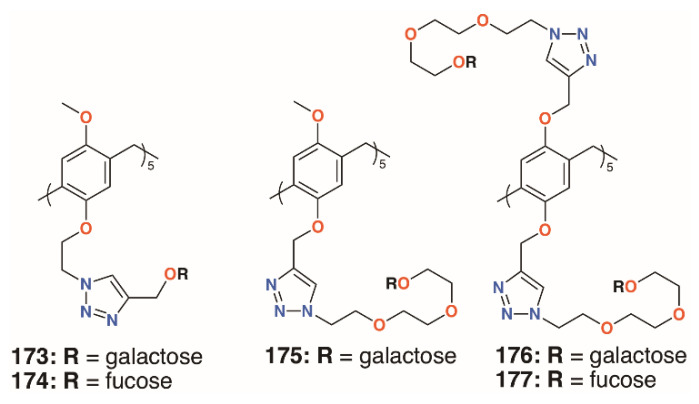
Vidal’s glycosylated pillar[5]arenes.

**Figure 29 molecules-25-05145-f029:**
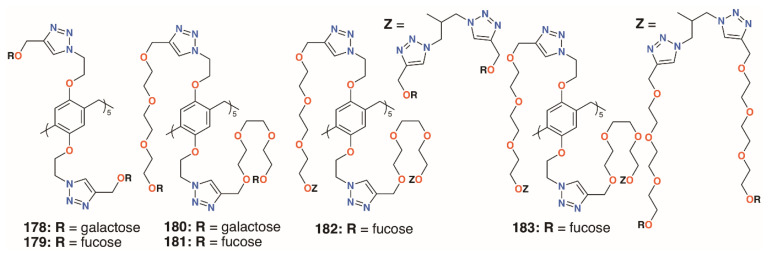
Nierengarten’s glycosylated pillararenes.

**Figure 30 molecules-25-05145-f030:**
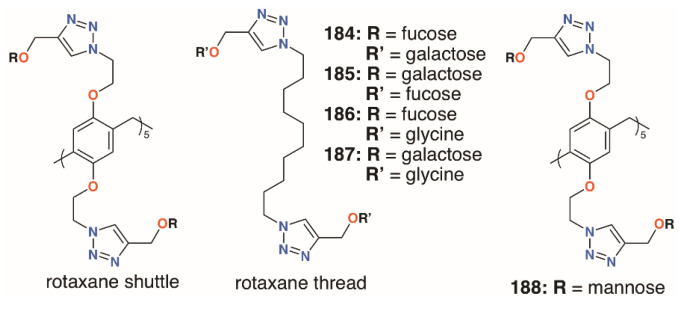
Nierengarten’s pillar[5]arene rotaxanes.

**Figure 31 molecules-25-05145-f031:**
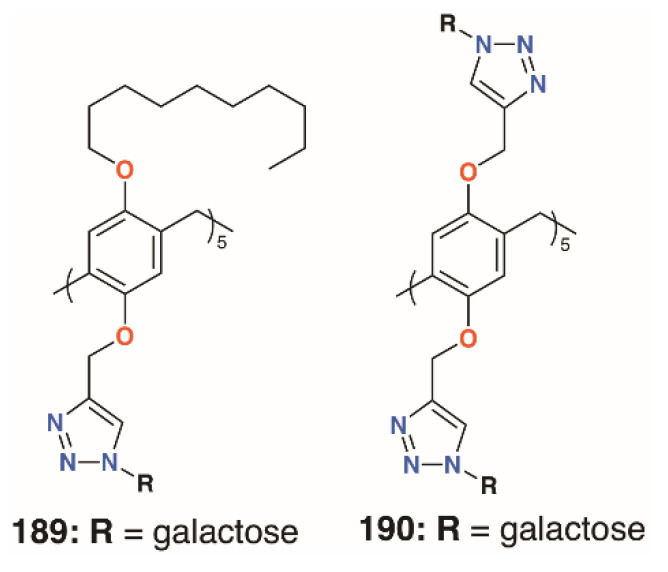
Huang’s glycosylated pillar[5]arenes.

**Figure 32 molecules-25-05145-f032:**
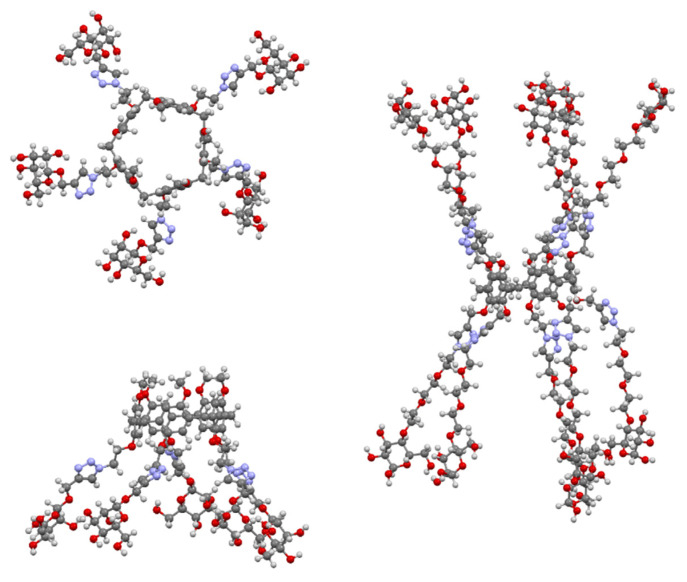
Glycosylated pillar[5]arenes **173** (**top** and **side**) and **176** (**side**) illustrating their multivalent binding potential.

**Figure 33 molecules-25-05145-f033:**
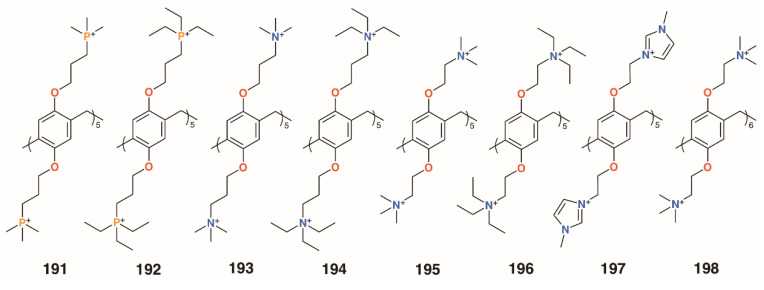
Cohen’s charged pillar[*n*]arenes.

**Table 1 molecules-25-05145-t001:** Summary of macrocycles with significant antibiotic activities. MIC, minimum inhibitory concentration.

Compound	Target	MIC/μg mL^−1^	Ref
**5**	*E. coli*	16	[20]
	*S. aureus*	16	[20]
	*E. faecium*	16	[20]
**6** (*cone*)	*E. faecalis*	8	[23]
	*M. tuberculosis*	9.5	[23]
**6** (*1,3-alt*)	*M. tuberculosis*	1.2	[23]
8	*E. coli*	4	[25]
	*S. aureus*	8	[25]
Fe·**47**	*E. coli*	0.37	[35]
	*S. albus*	0.37	[35]
	*R. stolonifera*	0.37	[35]
Cu_2_·**47**	*E. coli*	0.37	[36]
	*S. albus*	0.37	[36]
	*R. stolonifera*	0.75	[36]
**59**	*S. aureus*	7.8	[41]
	*S. epidermidis*	7.8	[41]
	*MRSA*	15.6	[41]
	*B. subtilis*	15.6	[41]
	*P. aeruginosa*	15.6	[41]
**60**	*S. aureus*	3.9	[41]
	*S. epidermidis*	15.6	[41]
	*MRSA*	0.97	[41]
	*B. subtilis*	0.97	[41]

**Table 2 molecules-25-05145-t002:** Summary of macrocycles with significant cytotoxic effects on bacterial and fungal species.

Compound	Target	MIC/μg mL^−1^	Ref
**5**	*M. tuberculosis*	1.22	[50]
**96**	*M. tuberculosis*	1.89	[50]
**105**	*S. aureus*	1.95	[53]
**109**	*S. aureus*	1.95	[55]
**122**	*S. aureus*	4	[59]
	*S. epidermidis*	16	[59]
**130**	*S. aureus*	3.05	[65]
	*S. epidermidis*	3.05	[65]
	*B. subtilis*	3.05	[65]
**131**	*S. aureus*	4.32	[65]
	*S. epidermidis*	4.32	[65]
	*B. subtilis*	4.32	[65]
**132**	*S. aureus*	5.31	[65]
	*S. epidermidis*	5.31	[65]
	*B. subtilis*	5.31	[65]
**133**	*S. aureus*	6.64	[65]
	*S. epidermidis*	6.64	[65]
	*B. subtilis*	6.64	[65]

**Table 3 molecules-25-05145-t003:** Summary of macrocycles demonstrating significant biofilm inhibition.

Compound	Target	MBIC/μg mL^−1^	Ref
**154**	*P. aeruginosa*	2	[73]
**155**	*P. aeruginosa*	1.1	[74]
**158**	*P. aeruginosa*	5.4	[76]
**165**	*S. aureus*	17.2	[80]
**168**	*L. pneumophila*	0.78	[82]
**168**·gatifloxacin	*S. aureus*	0.16	[82]
**169**	*S. aureus*	6.25	[83]
	*E. faecalis*	6.25	[83]
	*MRSA*	1.56	[83]
**172**·clarithromycin	*S. pneumoniae*	13.69	[85]
**191**	*S. aureus*	0.17	[92]
	*E. faecalis*	0.47	[92]
**192**	*S. aureus*	0.17	[92]
	*E. faecalis*	0.47	[92]
**193**	*S. aureus*	0.17	[92]
	*E. faecalis*	0.47	[92]
**194**	*S. aureus*	0.17	[92]
	*E. faecalis*	0.47	[92]
**195**	*S. aureus*	0.20	[92]
	*E. faecalis*	0.20	[92]
	*S. mutans*	0.50	[92]
**198**	*S. aureus*	0.11	[92]
	*E. faecalis*	0.11	[92]
	*S. epidermidis*	0.23	[92]

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
