# Peer review of "Antimicrobial Activity of Calixarenes and Related Macrocycles"

_molecules, 2020, doi:10.3390/molecules25215145_

Round 1
Reviewer 1 Report
This review article is a nice addition to the field of calixarene chemistry as its approach of focusing on the specific subject of anti-microbial activity has not been previously attempted. The authors adopt a good critical and analytical tone. The standard of preparation of the manuscript is very high with good quality diagrams and referencing
There are a few minor changes to be made before publication
- Subheadings would greatly help the reader
- General intro – the calixsugars have been reviewed both in journals and books – include a reference to this?
- Some general references are missing e.g. section 2.1 – one of the books on calixarenes and something to magic bullet. Latter on some references to biofilm biology.
- The work of Hamid on sulphonamides is considered under cation releasing calixarenes, but no mention are made of cations – its seems it should be in the earlier section with other pro-drugs
- The pillararene diagrams in the biofilm section lack clarity due to the side chains – could a 3D model (as in figure 1) be introduced?
Author Response
Reply to reviewers’ comments
Reviewer 1
Subheadings would greatly help the reader
General intro – the calixsugars have been reviewed both in journals and books – include a reference to this?
Two book chapters have been included in the references
Some general references are missing e.g. section 2.1 – one of the books on calixarenes and something to magic bullet. Later on some references to biofilm biology.
References to the most recent book on calixarenes, Calixarenes and Beyond, a paper on Ehrlich’s ‘magic bullet’ concept, and an excellent short review on biofilm formation have been added.
The work of Hamid on sulphonamides is considered under cation releasing calixarenes, but no mention are made of cations – its seems it should be in the earlier section with other pro-drugs.
This has been combined under a subheading with other cation-releasing calixarenes.
The pillararene diagrams in the biofilm section lack clarity due to the side chains – could a 3D model (as in figure 1) be introduced?
A figure, similar to Figure 1, now illustrates two such compounds.
Reviewer 2 Report
The review Antimicrobial Activity of Calixarenes and Related Macrocycles of Peter J. Cragg and co-workers is a useful addition on the potential biological application of the important class of macrocycles.
The Authors present in some details the Antimicrobial Activity of various calixarenes and related molecules reported in literature.
The review is well written and appears to majority of literature available, with some small exceptions (see below).
The review is suitable for publication in Molecules after some minor revision as detailed below.
- A summary table indicating the state of the art of the most promising results would make it easier for the reader to have a rapid overview of the field.
- The biological activity reported is not homogenous in terms of measurement units (mg/kg; microg/ml; micoM; etc.). For comparative proposes a homogenous presentation (for example in microM) should be applied. This should be obligatory for a review presenting a lot of data from the literature.
- Please check the sentences at the following lines: 67-69; 104-105; 208-210; 220; 295-297; 387-390; 456-459; 522-524; 584-585; 629-631.
- Fig 10 in line 208 should be Fig. 9
- The conclusions are quite brief and should include more detail on promising results.
- In the reference [5] the title is written in Cyrillic. Is it the correct reference? A more recent review can be found at the following link
- Please check also reference [46]
- The following relevant references should be added in the manuscript.
https://www.mdpi.com/1420-3049/25/2/370
https://pubs.acs.org/doi/abs/10.1021/acsomega.9b02948
https://www.sciencedirect.com/science/article/pii/S1631074802013541
https://onlinelibrary.wiley.com/doi/full/10.1111/cbdd.12818
https://www.sciencedirect.com/science/article/pii/S0378517318305969
Author Response
Reply to reviewers’ comments
Reviewer 2
A summary table indicating the state of the art of the most promising results would make it easier for the reader to have a rapid overview of the field.
We had intended to provide this but had a deadline to meet. In revising the manuscript, we have had time to compile three tables highlighting the most active compounds for which comparative MIC or MIBC data have been published.
The biological activity reported is not homogenous in terms of measurement units (mg/kg; microg/ml; micoM; etc.). For comparative proposes a homogenous presentation (for example in microM) should be applied. This should be obligatory for a review presenting a lot of data from the literature.
This has been done. Molar values would have been ideal, however, some substances, such as Macrocyclon, do not have a specific molecular mass. Furthermore, most biologists weigh out materials for MIC tests Consequently we have converted all units to mg ml-1.
Please check the sentences at the following lines: 67-69; 104-105; 208-210; 220; 295-297; 387-390; 456-459; 522-524; 584-585; 629-631.
These have been checked and amended as necessary.
Fig 10 in line 208 should be Fig. 9
Corrected.
The conclusions are quite brief and should include more detail on promising results.
A few lines identifying key themes, together with their likely future impact, have been added.
In the reference [5] the title is written in Cyrillic. Is it the correct reference? A more recent review can be found at the following link
The original reference is correct; this more recent review is on the broader subject of ‘Calixarene Derivatives for (Nano)Biotechnologies’. While checking this, we came across a few other relevant reviews so these have also been cited.
Please check also reference [46]
This has been checked and the English transliteration of the authors’ names have been improved.
The following relevant references should be added in the manuscript.
https://www.mdpi.com/1420-3049/25/2/370
https://pubs.acs.org/doi/abs/10.1021/acsomega.9b02948
https://www.sciencedirect.com/science/article/pii/S1631074802013541
https://onlinelibrary.wiley.com/doi/full/10.1111/cbdd.12818
https://www.sciencedirect.com/science/article/pii/S0378517318305969
The first four have been included. We were aware of the fifth paper but, on reading it (and its supplementary information containing the compounds’ synthesis), the structures of these complex derivatives remained unclear as they are never illustrated. From the paper it appeared that the modified calixarenes were not significantly more potent that the drug molecules that were attached to them. Consequently it was decided to omit this reference.
Reviewer 3 Report
This manuscript is a review on calixarenes with antimicrobial activity. The need of such a review is well justified because, as the authors state, the antimicrobial activity of calixarenes has only been reviewed in the Ukrainian literature in 2015.
The organization of the information in this review is appropriate as well as the papers that have been included. Apart from this, I have appreciated that the authors make a short general introduction at the beginning of the sections, which may be helpful to the readers. For all these reasons, I believe that this review would be of great interest to scientists working in this domain.
However, there are two aspects that should be improved and that are the reason for which this paper needs major revisions:
- The chemical structure is as important as the description of the biological activity of the compounds. In many cases, the description of the structure is not correct and should be revised. In other cases, it should be extended.
For instance:
- Lines 98 - 100: “an analogue, 2, in which penicillin V was appended to the calixarene [10] and tested against Gram-positive and Gram-99 negative bacteria [11]. The group also prepared a nalidixic acid delivering prodrug, 3.”. However, this description does not match the structures of 2 and 3 in Figure 3.
- Lines 163 and 164: “4-t-butylcalix[4]arenes, 9 to 17, with 1,3,4-oxadiazole and 1,3,4-thiadiazole derivatives of isoniazid, nicotinic acid, benzoic acid, and cis-164 cinnamic acid (Figure 5)”. However, in Figure 5, these compounds do not possess a heterocycle with 3 heteroatoms. The heterocycle only has one heteroatom.
- The structure of compounds 31 to 35 could be described in the text.
- Line 214: is the “O-butylsulfonate” correct? Compounds 41 to 46 do not have a butyl substituent.
- More details on the structure of diamides 48 to 51 could be included in the text.
- Lines 346 – 347: among compounds 88 to 99, some of them bear a bithiazolyl moiety. This should be clarified in the text.
- Line 346: why these compounds are named 6,6’-dimethyl-2,2’-bipyridyl? In one of the pyridines, I am not able to find a methyl group at position 6.
- Line 354: please revise “2-hydroxy-N,N,N–dimethylmethanaminium”.
- Lines 353 – 356 and line 371: the structure of compounds 104 and 105 should be described in order to differentiate it from that of the other compounds in figure 16.
- Figure 20: the structure of compound 128 is missing.
These are only some examples. The authors should thoroughly revise the entire manuscript.
- The titles “2. Mode of action” and “2.3. Cell destruction” are not appropriate. In fact, section 2 does not describe the mode of action of the compounds (papers on the mechanism of action are not included). In general, it describes compounds with antimicrobial activity or antibiofilm activity.
Minor considerations:
- Line 208: “Figure 10” should be “Figure 9”.
- Line 214: “sulfaontocalix” should be corrected.
- Line 266: “chloramphicol” should be corrected.
- The grammar of some sentences should be revised.
In fact, the authors should thoroughly revise the spelling of the entire manuscript.
Author Response
Reply to reviewers’ comments
Reviewer 3
The chemical structure is as important as the description of the biological activity of the compounds. In many cases, the description of the structure is not correct and should be revised. In other cases, it should be extended.
On reviewing the structures, we agree with the reviewer and have corrected a number of structures.
For instance:
Lines 98 - 100: “an analogue, 2, in which penicillin V was appended to the calixarene [10] and tested against Gram-positive and Gram-99 negative bacteria [11]. The group also prepared a nalidixic acid delivering prodrug, 3.”. However, this description does not match the structures of 2 and 3 in Figure 3.
Structures 2 and 3 have been reversed in the figure and it is now correct.
Lines 163 and 164: “4-t-butylcalix[4]arenes, 9 to 17, with 1,3,4-oxadiazole and 1,3,4-thiadiazole derivatives of isoniazid, nicotinic acid, benzoic acid, and cis-164 cinnamic acid (Figure 5)”. However, in Figure 5, these compounds do not possess a heterocycle with 3 heteroatoms. The heterocycle only has one heteroatom.
Two nitrogen atoms had been omitted from the structures. The compounds now contain heterocycles with three heteroatoms.
The structure of compounds 31 to 35 could be described in the text.
This has been done.
Line 214: is the “O-butylsulfonate” correct? Compounds 41 to 46 do not have a butyl substituent.
The text named calix[6]- and [8]sulfonates, originally compounds 45 and 46, as having butylsulfonate substituents. The text has been corrected to ‘propysulfonate’.
More details on the structure of diamides 48 to 51 could be included in the text.
This has been done.
Lines 346 – 347: among compounds 88 to 99, some of them bear a bithiazolyl moiety. This should be clarified in the text.
This has been clarified.
Line 346: why these compounds are named 6,6’-dimethyl-2,2’-bipyridyl? In one of the pyridines, I am not able to find a methyl group at position 6.
The names for these compounds have been simplified to avoid incorrect nomenclature which had come from the original paper.
Line 354: please revise “2-hydroxy-N,N,N–dimethylmethanaminium”.
This has been revised.
Lines 353 – 356 and line 371: the structure of compounds 104 and 105 should be described in order to differentiate it from that of the other compounds in figure 16.
This has been done.
Figure 20: the structure of compound 128 is missing.
The number 127 had been duplicated; the second derivative should have been 128. This has been corrected.
These are only some examples. The authors should thoroughly revise the entire manuscript.
A thorough revision of the manuscript has been done.
The titles “2. Mode of action” and “2.3. Cell destruction” are not appropriate. In fact, section 2 does not describe the mode of action of the compounds (papers on the mechanism of action are not included). In general, it describes compounds with antimicrobial activity or antibiofilm activity.
The headings and subheadings have been changed to more accurately reflect the compounds described.
Minor considerations:
Line 208: “Figure 10” should be “Figure 9”.
Line 214: “sulfaontocalix” should be corrected.
Line 266: “chloramphicol” should be corrected.
These have been corrected.
The grammar of some sentences should be revised.
This has been done.
In fact, the authors should thoroughly revise the spelling of the entire manuscript.
The manuscript has been thoroughly revised to ensure consistent US English spelling throughout.
Round 2
Reviewer 3 Report
The manuscript has been revised according to my suggestions.